# A cysteine-less and ultra-fast split intein rationally engineered from being aggregation-prone to highly efficient in protein *trans*-splicing

Christoph Humberg [ID] , Zahide Yilmaz, Katharina Fitzian [ID] , Wolfgang Dörner [ID] , Daniel Kümmel [ID] & Henning D. Mootz [ID] [✉]

Split inteins catalyze protein *trans*-splicing by ligating their extein sequences while undergoing self-excision, enabling diverse protein modification applications. However, many purified split intein precursors exhibit partial or no splicing activity for unknown reasons. The Aes123 PolB1 intein, a representative of the rare cysteine-less split inteins, is of particular interest due to its resistance to oxidative conditions and orthogonality to thiol chemistries. In this work, we identify β-sheet-dominated aggregation of its N-terminal intein fragment as the origin of its low (~30%) splicing efficiency. Using computational, biochemical, and biophysical analyses, we characterize the fully active monomeric fraction and pinpoint aggregation-prone regions. Supported by a crystal structure, we design stably monomeric mutants with nearly complete splicing activity. The optimized CLm intein (Cysteine-Less and monomeric) retains the wild-type's ultra-fast reaction rate and serves as an efficient, thiol-independent protein modification tool. We find that other benchmark split inteins show similar precursor aggregation, suggesting that this general phenomenon arises from the intrinsic challenge to maintain the precursor in a partially disordered state while promoting stable folding upon fragment association.

Inteins are autoprocessing domains that mediate the virtually traceless ligation of their flanking extein polypeptide sequences in a precursor protein with their own concomitant removal[1,2]. This process is termed protein splicing and encompasses a coordinated multistep rearrangement of the polypeptide backbone (Fig. 1 and Supplementary Fig. 1)[3–7]. It typically involves two thioester or oxyester intermediates on Cys, Ser or Thr side chains at the two splice junctions between exteins and intein. In split inteins two separate precursor proteins with $Int^N$ and $Int^C$ fragments, $P_N$ and $P_C$, first associate and fold into the active intein domain before undergoing a mechanistically analogous reaction termed protein *trans*-splicing (PTS; Fig. 1 and Supplementary Fig. 1)[4–6,8].

Inteins enable many protein engineering technologies that rely on their unique making and breaking of peptide bonds[7,9–11]. Split inteins are of particular potential for protein labeling, semi-synthesis and reconstitution applications as they assemble a peptide backbone from two separate proteins that are individually producible and modifiable.

Cysteine-less split inteins constitute a rare subgroup of inteins with additional potential as protein engineering tools[12,13]. They employ only oxyester intermediates with their serine or threonine residues at the splice junctions (Fig. 1)[14]. Ideally, also the remainder of the intein sequences contains no or only non-essential cysteines that can be removed by mutagenesis. Therefore, cysteine-less split inteins are fully active without the addition of reducing agents typically applied for

Institute of Biochemistry, University of Münster, Corrensstraße 36, 48149 Münster, Germany. [✉]e-mail: Henning.Mootz@uni-muenster.de

**Fig. 1 | Split intein-mediated protein *trans*-splicing.** Highlighted are the key residues at both splice junctions. The first residue of the Int^N fragment and the first residue of the Ext^C harbor nucleophilic side chains (Cys, Ser, Thr), which are only Ser or Thr in cysteine-less inteins. The inlets illustrate the folding states of the intein fragments of an intein split at the endonuclease position. These states change from molten globule (N1 lobe) and disordered (N2 lobe) in the Int^N fragment and disordered in the Int^C fragment prior to the electrostatically driven association to the folded and intertwined complex in the assembled intein. Ext^N, Ext^C = N- and C-terminal exteins; P_N, P_C = N- and C-terminal precursor; SP splice product.

cysteine-dependent inteins to prevent oxidative inactivation. Owing to the insensitivity to oxidative conditions and their orthogonality with respect to thiol-directed bioconjugation reagents, cysteine-less split inteins enable attractive new schemes that exploit thiol chemistry in the extein sequences or involve PTS in the oxidative extracellular milieu, for example[12].

Importantly, however, a cysteine-less split intein with a high efficiency, high rate and high affinity of the precursor fragments is still lacking. These are the most important activity traits for all split inteins to enable the effective manipulation of proteins both in purified form and in living cells. The cysteine-less Aes123 PolB1 intein (in short: Aes intein) found inserted in PolB-type DNA polymerase genes from T4-like bacteriophages of *Aeromonas salmonicida* shows only about 30% splicing efficiency relative to its P_N[12]. While the splicing efficiency is significantly increased by shifting the split site (see below), the resulting engineered split intein exhibits a strongly reduced rate in PTS[12].

In general, only a few known split inteins exhibit high performance in terms of efficiency, rate and fragment affinity. Currently, these properties cannot be predicted solely from sequence data and require an elaborate biochemical and biophysical characterization for each individual split intein. Naturally split inteins are considered to be more active than those that are artificially generated by genetically splitting a *cis*-splicing intein within its conserved horseshoe fold of about 130–150 aa[8,15–17], since the latter lack evolutionary adaptation to

the split form. The most common split site found in nature, also used for artificially split inteins, coincides with a position where most *cis*-inteins harbor an insertion of a homing endonuclease domain (Supplementary Fig. 1a)[18,19], resulting in Int^N and Int^C fragments of about 100 aa and 40 aa, respectively. Examples are the high performing and widely used, naturally split *Npu* DnaE and Gp41-1 inteins[20–22]. In contrast, atypically split inteins operate with a short Int^N fragment of about 15 to 25 aa and a proportionally longer Int^C fragment[23–25]. However, also naturally split inteins often exhibit poor efficiencies for unknown reasons[21,26] and a wide range of rates with half-life times from seconds to hours[21,22,27]. Many inteins that are artificially split at the endonuclease position show good efficiency when the P_N and P_C precursors are co-expressed in cells[16,17]. However, when the precursors are separately purified before their combination, the efficiencies are typically limited to 40-50% at best[28–30]. Nearly inactive purified precursors of some artificially split intein precursors are reported to be capable of splicing when collectively refolded from denaturing conditions[15,31–33], suggesting folding issues as underlying reason. In contrast, artificially splitting inteins at the atypical N-terminal position can result in high splicing efficiency of the purified precursors, as mentioned above for the Aes intein[12,34]. Overall, the molecular origins for incomplete PTS appear to be associated with precursor (mis-)folding but remain unknown to date.

Folding of the intein precursors split at the endonuclease position into the conserved and intertwined horseshoe structure of inteins is initiated by electrostatic association (capture) of disordered regions within each fragment with opposite charges[35,36]. The collapse into the conserved and intertwined horseshoe structure of inteins then triggers PTS (Fig. 1). The transition from the disordered to the ordered state is thermodynamically favorable through the formation of a structure rich in β-strands[35,36]. For the *Npu* DnaE intein, the opposite charges have been mapped to disordered regions in the C-terminal of the two symmetry-related lobes (N2) within the Int^N fragment, and to the Int^C fragment. In contrast, the N-terminal lobe of the Int^N fragment (N1) exhibits a residual, molten-globule-like structure before fragment association (Fig. 1)[36].

In this work, we revisit the Aes intein to understand the molecular origins for its limited efficiency. We reveal that a highly aggregated, yet soluble form of the P_N causes the partial inactivity. We identify regions in the disordered part of the Int^N that are prone to aggregation and obtain a fully monomeric variant with nearly complete efficiency by rational mutagenesis. Combined with its ultra-fast rate, the engineered Aes intein (CLm intein) is a highly valuable addition to the protein chemistry toolbox. We further show that the formation of soluble aggregates is a more widely occurring feature of both artificially and naturally split inteins, thus providing an explanation for a long-standing bottleneck in intein research.

## Results

### The active fraction of the cysteine-less Aes split intein splices at an ultra-fast rate

We revisited the naturally occurring Aes split intein with its Aes^N (120 aa) and Aes^C (39 aa) fragments as a detailed biochemical characterization was abandoned in a previous study due to its poor activity[12]. We prepared recombinant model N- and C-terminal precursor proteins MBP-Aes^N-H_6 (**1P**; P_N), Aes^C-GFP (**2P**; P_C) and SBP-Aes^C-SBP (**3P**; P_C) with maltose-binding protein (MBP), green fluorescent protein (GFP) or streptavidin-binding peptide (SBP) as exteins, respectively (Fig. 2a). Three native flanking residues were kept on each side of the intein fragments here and throughout this work unless stated differently (DTD and SVY, respectively). Proteins were produced in *Escherichia coli* and purified using Ni-NTA or streptactin affinity chromatography. In the protein samples and assays no reducing agents were applied. Upon mixing complementary constructs **1P** and **2P** in a 1:3 ratio at 37 °C, we observed splice product formation of about 30%, relative to

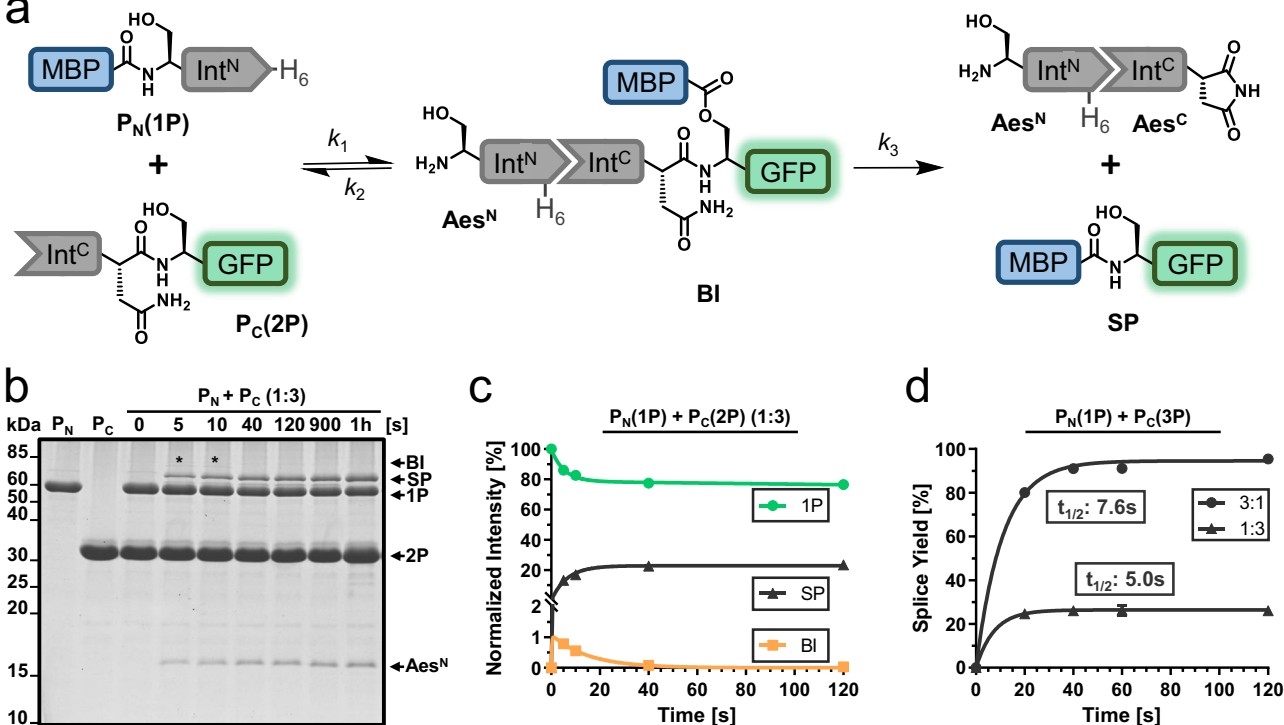

**Fig. 2 | Protein *trans*-splicing of the wildtype Aes intein. a** Scheme of the PTS reaction in a simplified three-state kinetic model. The Aes$^N$ and Aes$^C$ fragments are 120 and 39 aa in size, respectively. Not shown are the linear intermediate resulting from the *N-O* acyl shift and the C-terminal succinimide prior to hydrolysis. **b** SDS-PAGE analysis of the PTS reaction at 37 °C using precursors **1P** and **2P** at 10 µM and 30 µM, respectively, and in the absence of any reducing agents. This experiment was repeated three times. **c** Time-course of the PTS reaction obtained by densitometric analysis of data as shown in (**b**) and three-state kinetic model fitted to the data. **d** Splice product formation plotted against the time applying different molar ratios of the precursors **1P** and **3P**. Calculated molecular masses are: $P_N(1P) = 58.2$ kDa, $P_C(2P) = 31.3$ kDa, BI = 74.8 kDa, SP = 70.1 kDa, Aes$^N$ = 14.7 kDa, Aes$^C$ = 4.7 kDa. BI branched intermediate, SP splice product. For (**c**, **d**), $n = 3$ technical replicates. Data are presented as mean ± s.d. normalized to the molecular weight of each protein species. Source data are provided as a Source Data file.

the limiting Aes$^N$ precursor **1P** (Fig. 2b). These results confirmed the intein's poor efficiency[12], which significantly limits its utility.

Interestingly, however, the reaction showed an ultra-fast rate of $138.7 \pm 18.6 \times 10^{-3}\,s^{-1}$ ($t_{1/2} = 5.0\,s$) at 37 °C (Fig. 2c). A three-fold molar excess of the Aes$^N$ precursor led to virtually quantitative splicing of the Aes$^C$ precursor (**1P + 3P**) with a rate of $91.3 \pm 6.2 \times 10^{-3}\,s^{-1}$ ($t_{1/2} = 7.6\,s$) (Fig. 2d, Supplementary Fig. 2). N- or C-cleavage as potential side reactions (see Supplementary Fig. 1b) were below the detection limit. This ultra-fast splicing rate of the Aes intein was previously overlooked and ranks this intein among the fastest known, e.g. the Gp41-1 intein ($t_{1/2} = 5\,s$ at 37 °C)[22,37] and the NrdJ-1 intein ($t_{1/2} = 7\,s$)[22], and more than 10-fold faster than the *Npu* DnaE intein ($t_{1/2} = 63\,s$)[21], all of which are cysteine dependent.

**Concentration-dependent aggregate formation of the Aes$^N$ precursor is yield-limiting**

To address the substantial inactivity of the Aes$^N$ precursor we hypothesized it partially misfolds in the absence of its $P_C$ partner. To test co-folding of both $P_N$ and $P_C$, we denatured separately purified **1P** and **3P** with 8 M urea and then mixed both proteins in a 1:3 molar ratio (10 µM **1P** + 30 µM **3P**), followed by removal of the denaturant by dialysis. Indeed, this procedure led to virtually complete conversion of **1P** into splice product (Supplementary Fig. 3).

To investigate $P_N$ conformation in solution, we analyzed **1P** ($m_{calc} = 58.2\,kDa$) by size exclusion chromatography (SEC) (Fig. 3a, b). The protein appeared in two well-separated fractions. Fraction A (about 71% of the protein) eluted at the void volume of the column, thus corresponding to an apparent molecular weight (MW) of at least ca. 2000 kDa, indicative of a highly oligomeric or aggregated form of

**1P**. The retention time of fraction B (about 29%) translated into a Stokes radius of 31.2 Å, which reflects an apparent MW of about 64 kDa, in good consistency with a monomeric and partially disordered form. No smaller oligomeric species of dimers or trimers, etc. were detected. Most importantly, the aggregated fraction of **1P** showed only traces of PTS activity with an excess of $P_C$ **3P**, while the monomeric fraction spliced virtually quantitatively under these conditions (Fig. 3c). These findings explained the partial activity of the Aes$^N$ precursor. Variation of the N-extein to GFP in construct GFP-Aes$^N$-H$_6$ (**4P**) did not change the occurrence of the aggregated fraction, suggesting that aggregation is an inherent property of the Aes$^N$ fragment (Supplementary Fig. 4).

Attempts to prevent the aggregated fraction by varying the expression conditions or applying different salt concentrations failed (Supplementary Fig. 4). We therefore developed a refolding protocol to obtain monomeric $P_N$, for further analysis. Following Ni-NTA purification under denaturing conditions, **1P** and **4P** were stepwise refolded by dialysis in the presence of sucrose as a stabilizer. Once prepared this way, both proteins exhibited nearly quantitative PTS activity and retained ultra-fast rates with $t_{1/2} = 7.7$ and $9.5\,s$ with partner protein SBP-Aes$^C$-SBP (**3P**) (Fig. 3d, Supplementary Fig. 4). Using biolayer interferometry, we investigated the binding affinity of the monomeric split precursors inactivated by mutations to block splicing (Supplementary Fig. 5). A high affinity with a $K_d$ value of 12 nM with a $k_{on}$ of $4.9 \pm 1.8 \times 10^4\,M^{-1}s^{-1}$ and $k_{off}$ of $7.5 \pm 1.1 \times 10^{-4}\,s^{-1}$ revealed an additional favorable property of the split Aes intein next to its rate and cysteine-less nature. Assuming splice competence of the intein fragments, simulation of the underlying binding kinetics even indicated a $K_d$ of 0.5 nM (Supplementary Fig. 6, Supplementary Tab. 3). Furthermore,

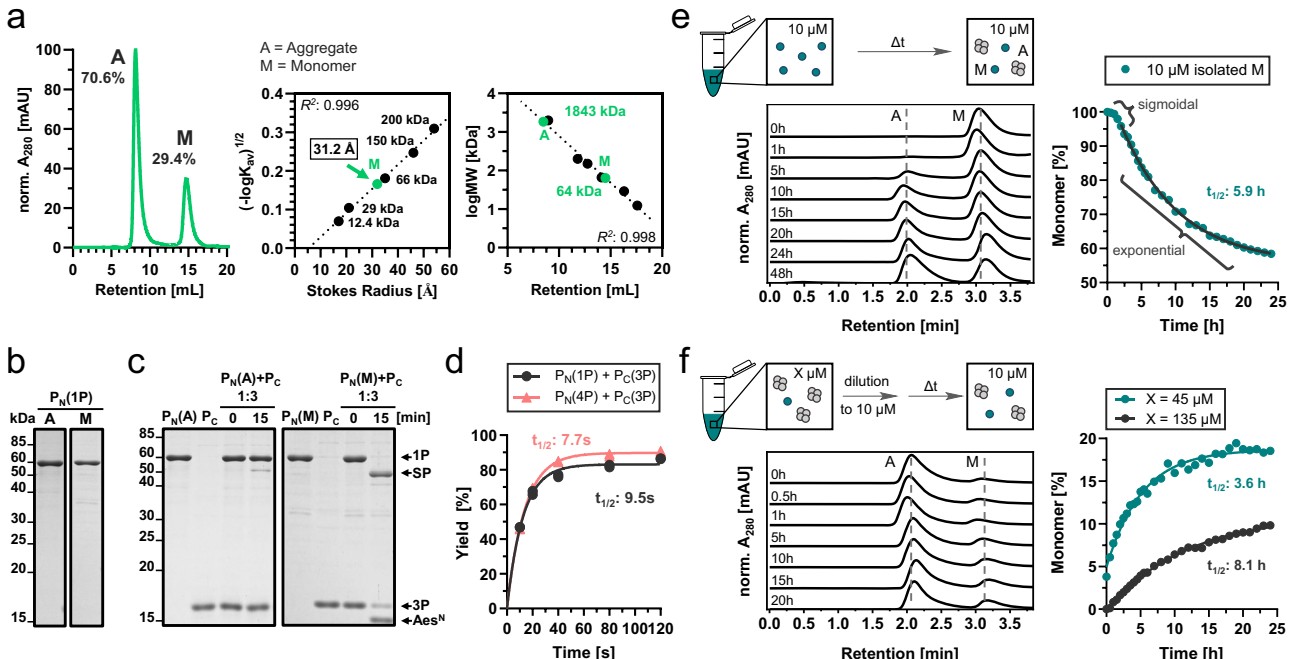

**Fig. 3 | The Aes^N precursor 1P exhibits two folding states as revealed by SEC. a** SEC analysis of **1P** (left panel). This data and a calibration curve generated with globular protein standards were used to determine the Stokes radius (middle panel) and the apparent molecular weight of the aggregated (A) and monomeric (M) species. Because the aggregated species eluted shortly before dextran as the void volume marker, we could only estimate its hydrodynamic radius to be >270 Å. **b** Analysis of aggregated and monomeric fractions on a Coomassie-stained SDS-PAGE gel. **c** Analysis of PTS reactions using the aggregated and monomeric species of **1P** (10 µM) with three-fold excess of **3P** (30 µM) at 37 °C. Shown are Coomassie-stained SDS-PAGE gels. **d** Time-courses of two PTS reactions using monomeric fractions of the indicated Aes^N precursors **1P** and **4P** with a three-fold excess of **3P** (see also Supplementary Fig. 4). **e** Analysis of concentration-dependent re-aggregation of **1P** over time. The left panel shows representative analytical SEC profiles recorded at indicated time points following the isolation of its monomeric fraction adjusted to 10 µM. The right panel shows the plot of the monomeric fraction. Note that only the second, exponential phase was used for fitting. The sigmoidal appearance of the curve at early time points is expected as a typical feature of amyloid aggregate formation due to a slow nucleation (lag phase) before aggregation proceeds as a concentration-dependent polymerization reaction (exponential phase) until reaching saturation (equilibrium phase)[46]. **f** Analysis of concentration-dependent resolution of **1P** aggregate. The left panel shows representative analytical SEC profiles recorded at indicated time points after dilution of an aggregate fraction from 45 µM to 10 µM. The right panel shows the plotted aggregate resolution to the monomeric species from starting concentrations 135 µM and 45 µM, each fitted using a one-phase exponential equation. For (**a**) (middle and right inlets), *n* = 3 technical replicates. Data are presented as mean. Source data are provided as a Source Data file.

the binding data revealed biphasic behavior consistent with the assumed split intein assembly mechanism (see Supplementary Note 1).

However, we then discovered that the elaborate refolding protocol of the active, monomeric P_N fractions was foiled as a useful preparative procedure by the tendency of the P_N to re-aggregate and form a monomer-aggregate ratio in a concentration-dependent manner (Fig. 3e). The ~30% of monomeric form of **1P** described so far was observed for an isolated protein concentration of ~10–15 µM, while at 45 µM and 135 µM only about 4% and no detectable monomeric species were present, respectively (Fig. 3f). Consequently, protein dilution induced disaggregation, however, at inconvenient concentrations and time scales. Further work with the wildtype intein was thus impractical and engineered Aes split intein variants resistant to aggregation were required.

### Disordered regions of the Int^N N2 and N1 lobes are drivers for β-sheet-dependent aggregate formation

To better understand the origins of the Aes^N aggregation, we carried out computational, biophysical and biochemical analyseis. Sequence analysis confirmed the typical opposite charges of the Aes^N and Aes^C pieces. The charge/hydrophobicity distribution indicated a partial folding of the first lobe (N1) of the Aes^N as well as both order promoting and disorder promoting characteristics in the second lobe N2 of the Aes^N and in the Aes^C fragment (Supplementary Fig. 7). We corroborated the overall only partially folded and largely disordered structure of the Aes^N monomer by circular dichroism (CD) spectroscopy and

thermal shift analysis using an inactivated Aes^N precursor with a minimal N-extein (residues DTD) (Fig. 4a, Supplementary Fig. 8, 9). Importantly, CD spectroscopy also revealed that the aggregated form of P_N is dominated by β-sheets. To closer pinpoint disordered regions within the Aes^N monomer we performed carbene footprinting (Fig. 4b, Supplementary Fig. 10). A folded or partially folded protein is assumed to be better shielded against labeling with a reactive carbene species than a disordered region. Carbene-labeling of a monomeric Aes^N fraction, followed by trypsinization and quantitative ESI-MS analysis showed that three tryptic-peptides (Pep4 to Pep6) identified from the second half of the Aes^N sequence were more prone to labeling than the three peptides identified from the first half (Pep1 to Pep3) (Fig. 4c). Notably, however, with respect to the two symmetry-related N1 and N2 lobes, Pep4 (aa61-67) belongs to the N-terminal lobe N1, considering the symmetry-axis between N1 and N2 at aa68-69. All six peptides were significantly more protected against carbene-labeling in a folded Aes^N–Aes^C complex, as expected (Fig. 4c, Supplementary Fig. 10).

To further investigate which parts of the Aes^N were the drivers for the aggregation we separately produced two N-terminal (aa1-61 (**11P**) and aa1-68 (**12P**)) and two C-terminal (aa62-120 (**13P**) and aa69-120 (**14P**)) segments according to the carbene-labeling results. We used SEC to analyze these segments as fusion proteins with MBP for their aggregation tendency (Fig. 4d). Only **13P**, the only construct encompassing all three peptides Pep4, Pep5 and Pep6, showed a significant aggregated fraction (53%). Segment **14P**, encompassing the two

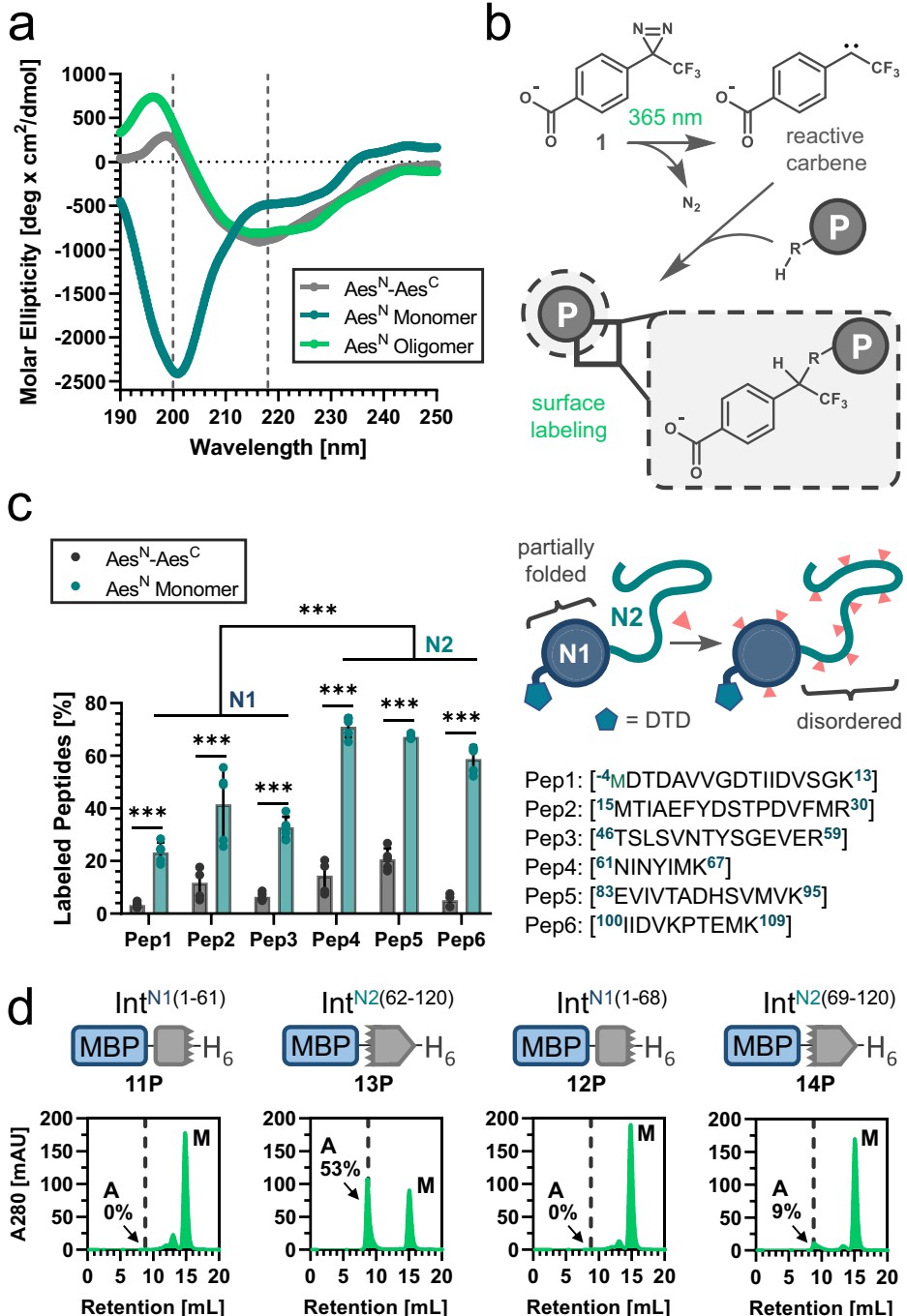

**Fig. 4 | Mapping of folding states within the Aes^N precursor. a** Far UV circular dichroism spectroscopy of the monomeric (blue) and the aggregated (green) species of DTD-(S1A)Aes^N (**8P**) in comparison to the fused complex structure MYIDTD-Aes^N(S1A)-GSH-Aes^C(N159A)-SVYLN (**9P**) (gray). **b** Scheme of the carbene-footprinting reaction to label exposed protein regions using trifluoromethylaryl diazirine (**1**). **c** MS-analysis of carbene labeling in Aes^N tryptic peptides (Pep1–Pep6) in the monomeric form of **8P** and the complex structure of MDTD-Aes^N(S1A)-GSH-Aes^C(N159A)-SVYLN (**10P**; 10 μM each) after 2 s of UV irradiation at 77 K using

10 mM aryldiazirine. **d** SEC-analysis of aggregate and monomer species content of protein constructs (30 μM each) containing partial Aes^N fragments. For (**a**), $n = 3$ technical replicates. Data are presented as mean molar ellipticity corrected for concentration. For (**c**), $n = 5$ technical replicates. Data are presented as mean ± s.d. normalized to the total peptide intensity. $p$-values are derived from a two-way ANOVA Sidak's multiple comparison test (***$p$ = < 0.0001). Source data are provided as a Source Data file.

C-terminal Pep5 and Pep6, showed little aggregated species that increased at higher concentration (9% at 30 μM and 15% at 150 μM; Supplementary Fig. 11). Together, not only the N2 lobe, but also parts of the N1 lobe seemed to exhibit a more disordered structure than the remainder of the Aes^N precursor, and these regions seemed to be responsible for the formation of soluble aggregates.

## Design of aggregation-reducing mutations that retain intein activity

We noted that the behavior of the Aes^N fragment was reminiscent of β-sheet-rich amyloid-like fibrils that require an at least partially disordered region with aggregation-prone sites to form the initial fibril nucleus[38,39]. We hypothesized that the identification of such nucleation

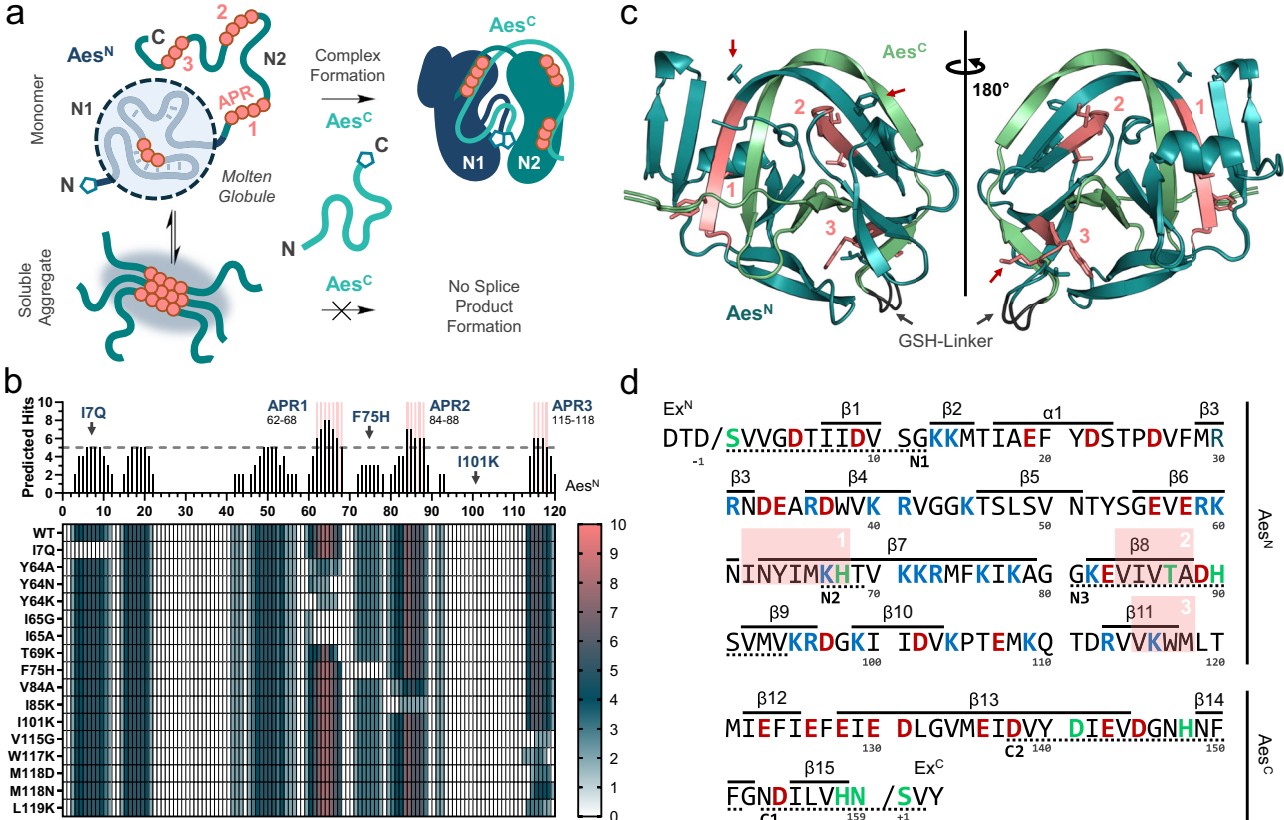

**Fig. 5 | Selection of mutations for the Aes^N engineering strategy. a** Model of Aes^N aggregation. Unsaturated aggregation-prone regions (APRs) in disordered regions of Aes^N serve as nuclei for soluble aggregate formation. **b** Sequence-based consensus prediction of amyloidogenic pattern formation in the Aes^N fragment using the web tool AMYLPRED2 including 10 different prediction algorithms[40]. The upper graph indicates the number of positive hits of the different algorithms plotted against the native Aes^N fragment sequence. APRs labeled in red are defined by ≥5 positive hits (dashed line indicates threshold). The heat map below shows the effect of the selected single mutations on the same aggregation-prone site prediction.

**c** X-ray structure of the Aes intein, obtained with the fused protein MYIDTD-Aes^N(S1A)-GSH-Aes^C(N159A)-SVYLN (**9P**). Shown is an overlay of chain A and B in the unit cell (for details see Supplementary Table 4). The three strongest APRs are marked in the structure (red). The artificial GSH inker is shown in dark gray. Residues T69, F75 and M118 are indicated with red arrows. **d** Primary sequence of the Aes^N (aa1–120) and the Aes^C fragment (aa121-139), each with 3 extein residues. Indicated are positively (red) and negatively (blue) charged residues, conserved motifs (underlined), key catalytic amino acids (green) and the location of secondary structure elements. For panel **b**, source data are provided as a Source Data file.

sites within the disordered part of the Aes^N precursor should provide amino acid candidates for a mutational strategy to increase the monomeric and active fraction (Fig. 5a). Importantly, however, such mutations would have to maintain the association, cooperative folding and splicing activity with the Aes^C counterpart.

We used the bioinformatic web tool AMYLPRED2 to analyze the Aes^N sequence. This program integrates ten sequence-based prediction tools for amyloidogenic determinants that operate on different algorithms[40]. Fig. 5b shows a plot of the number of positive predictions along the Aes^N sequence. The more of these tools predicted the same aggregation-prone region (APR), the more probable we assumed this region to be an aggregation hotspot. Interestingly, the three strongest APRs predicted were all located in the C-terminal segment of Aes^N (aa60-120, including Pep4) consistent with our biophysical analysis.

We then designed mutations within the predicted regions of the C-terminal half of Aes^N. To this end, we also solved the crystal structure of an Aes^N-Aes^C fusion at 1.38 Å resolution to confirm that selected residues indeed formed β-sheet structures and to choose sterically fitting substitutions (Supplementary Tab. 4 and Supplementary Fig. 12, 13). Further insights from the crystal structure are discussed in Supplementary Fig. 13 and Supplementary Note 2. The structure also confirmed the presence of the recently discovered motif NX histidine as a unique residue conserved in cysteine-less inteins (Supplementary

Fig. 14)[13]. Interestingly, two of the three strongest APRs within Aes^N (APR1 & APR3) are directly in contact with Aes^C (Fig. 5c). Amino acids were changed to ones with a lower propensity for forming β-sheets[41]. Mutations towards polar and in particular positively charged side chains were preferred to avoid counteracting the electrostatically driven association with the Aes^C precursor. Catalytically important and highly conserved residues were not altered (Fig. 5d). In this way we manually designed 14 single mutations at 10 positions. Figure 5b shows how each of these impacted the aggregation profile of the Aes^N sequence according to a re-analysis with AMYLPRED2.

To experimentally test these mutations, we introduced them individually into MBP-Aes^N-H_6 (**1P**). To exclude concentration-dependent artifacts, each mutant protein was set at 10 μM and incubated for 24 h at 15 °C prior to its analysis for aggregate formation by SEC (Fig. 6a). To our delight, the majority of the mutants showed an increased percentage of monomeric fraction, ranging from 24 to 71% compared to 19% measured for the WT (**1P**) under these conditions. Two control mutations outside the APRs of the C-terminal segment of Aes^N, otherwise designed by similar principles (I7Q, I101K), did not have a positive influence (Fig. 6a).

We then performed PTS assays with Aes^C-GFP (**2P**) using all 16 mutants of **1P**, however, without SEC separation of the aggregated fraction. Figure 6a shows that splice yields indeed increased for most mutants in an overall good correlation with the increased monomeric

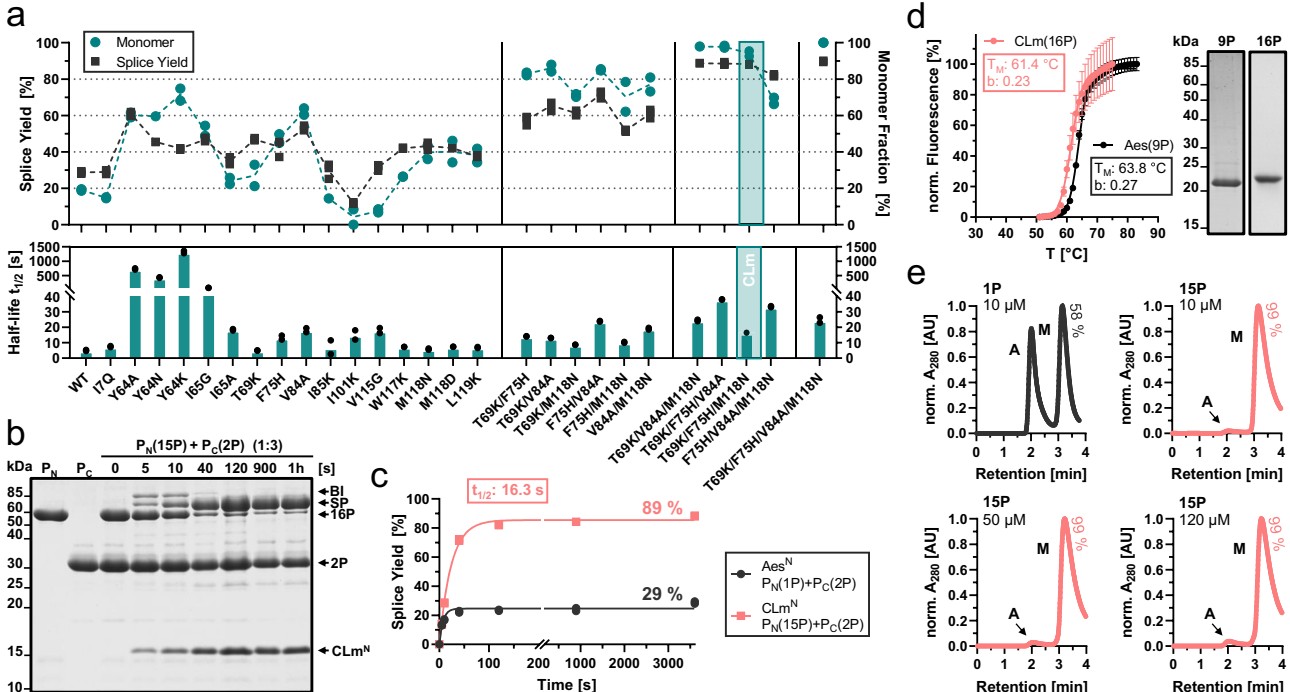

**Fig. 6 | Mutational engineering of monomeric and active Aes^N precursors.**
**a** Effects of the single mutations and selected mutual combinations introduced into
**1P**. The top panel shows splice yields (black) when mixed with three-fold excess of
the Aes^C partner **2P** (30 µM) for 1 h at 37 °C (see also Supplementary Fig. 16) and the
monomer proportion (blue) analyzed by SEC after incubation at 10 µM for 24 h at
15 °C. The lower panel shows the determined half-life times ($t_{1/2}$) of the PTS reaction. **b** Representative Coomassie-stained SDS-PAGE of the PTS reaction using the
CLm^N triple mutant **15P** (T69K/F75H/M118N) with three-fold molar excess of **2P**
(30 µM) at 37 °C. This experiment was repeated two times. **c** Time-courses of the
PTS reactions of the native MBP-Aes^N-H_6 (**1P**) and the corresponding CLm^N mutant

**15P** with three-fold molar excess of **2P** (30 µM) at 37 °C. **d** Thermal stability of the
fused CLm^N-Aes^C complex structure **16P** (red) compared to the fused Aes^N-Aes^C **9P**.
Indicated are the melting temperatures ($T_M$) and the Hill coefficient (b) of the
folding cooperativities. The Coomassie-stained SDS-PAGE gel (right panel) shows
the used constructs. **e** SEC analysis of re-aggregation of the SEC-purified mono-
meric Aes^N (**1P**) and CLm^N (**15P**) precursors after 24 h incubation at 15 °C at the
indicated concentrations. Aggregated (A) and monomeric (M) fractions are indi-
cated. BI = branched intermediate; SP = splice product. For (**d**), n = 5. Data are
presented as mean ± s.d. normalized to the highest fluorescence intensity. Source
data are provided as a Source Data file.

fraction (Pearson correlation coefficient ρ = 0.89; p = <0.0001). Their
splicing rates remained favorable and dropped at most only about
4-fold. Some mutants relatively underperformed in their splicing
ability (e.g. mutations at Y64) or showed increased levels of cleavage
reactions (e.g., **1P**(I65G)) and were not pursued further (Supplemen-
tary Fig. 15b).

Encouraged by these findings, we created double mutants of the 4
most promising single mutations (T69K, F75H, V84A and M118N) in all
six possible combinations. All double mutants showed further
increased percentages of the monomeric fraction and of the splice
yields, suggesting the individual mutations could act synergistically
(Fig. 6a). Furthermore, in terms of splicing rates they all remained in
the ultra-fast range with rate reductions of only 1.7 to 4.8-fold com-
pared to the WT. Double mutant **1P**(T69K/M118N) showed the best
performance in this regard with a $t_{1/2}$ = 8.7 s. We found the cleavage side
reactions slightly increased but still low in most cases (up to 1% and 3%
for C- and N-cleavage, respectively) (Supplementary Fig. 15c).

Subsequently, based on the same best single mutations, we pre-
pared all 4 possible triple mutants of MBP-Aes^N-H_6 (**1P**). Impressively,
these mutants formed monomeric species at >95%, except for
**1P**(F75H/V84A/M118N), and they showed virtually complete splicing
(about 90%). In the following, we focused on the mutant **1P**(T69K/
F75H/M118N), from now on termed CLm intein (cysteine-less mono-
meric; **15P**), as it retained the best ultra-fast splicing activity with
$t_{1/2}$ = 16.3 s, only about 3.3-fold slower than the WT (Fig. 6b, c), and
reached 89% splicing efficiency. We observed 3% of C-terminal clea-
vage side-product, but no detectable N-terminal cleavage (Supple-
mentary Fig. 15c). Notably, the negative impact of the T69K and F75H

mutations on the thermal stability of the fused Int^N-Int^C complex
(Supplementary Fig. 15a) was rescued in the CLm triple mutant, as the
melting temperature of the CLm^N-Aes^C complex **16P** (61.4 ± 0.2 °C)
was only slightly lower than for the WT Aes^N-Aes^C complex **9P** (Fig. 6d).
Most importantly, virtually no aggregated species could be detected
any more for the CLm^N precursor **15P**, consistent with its high splicing
efficiency. We also found that **15P** was resistant to re-aggregation over
a wide concentration range (Fig. 6e). Thus, this engineered CLm^N
precursor could be used for nearly quantitative and ultra-fast splicing
without any preceding SEC separation of the monomeric fraction.

Finally, we prepared the quadruple mutant **1P**(T69K/F75H/V84A/
M118N). While this mutant appeared even slightly more improved in
terms of monomeric behavior and splicing yield, its rate in PTS was
lower ($t_{1/2}$ = 24.7 s). For these reasons, we proceeded with the new CLm
mutant to explore splicing applications.

## The engineered CLm^N fragment shows extein generality and enhances expression yields

Unexpectedly, when testing the general utility of the CLm^N mutant for
various exteins, we noticed a positive influence on protein expression
yields. Compared to the WT Aes^N constructs, yields were improved 1.5
to 5.9-fold for different exteins and in different *E. coli* strains (Sup-
plementary Fig. 17a). For example, with MBP as N-extein the yield of
purified **15P** over **1P** was enhanced 1.6-fold. A construct with the ALFA
nanobody[42] (ALFAnb) as the N-extein, ALFAnb-CLm^N-H_6 (**17**), yielded
nearly 6-fold higher levels of purified protein. Furthermore, **17P**
exhibited virtually only monomeric and no aggregated species, com-
pared to only 45% monomeric species for **18P** containing the WT Aes^N

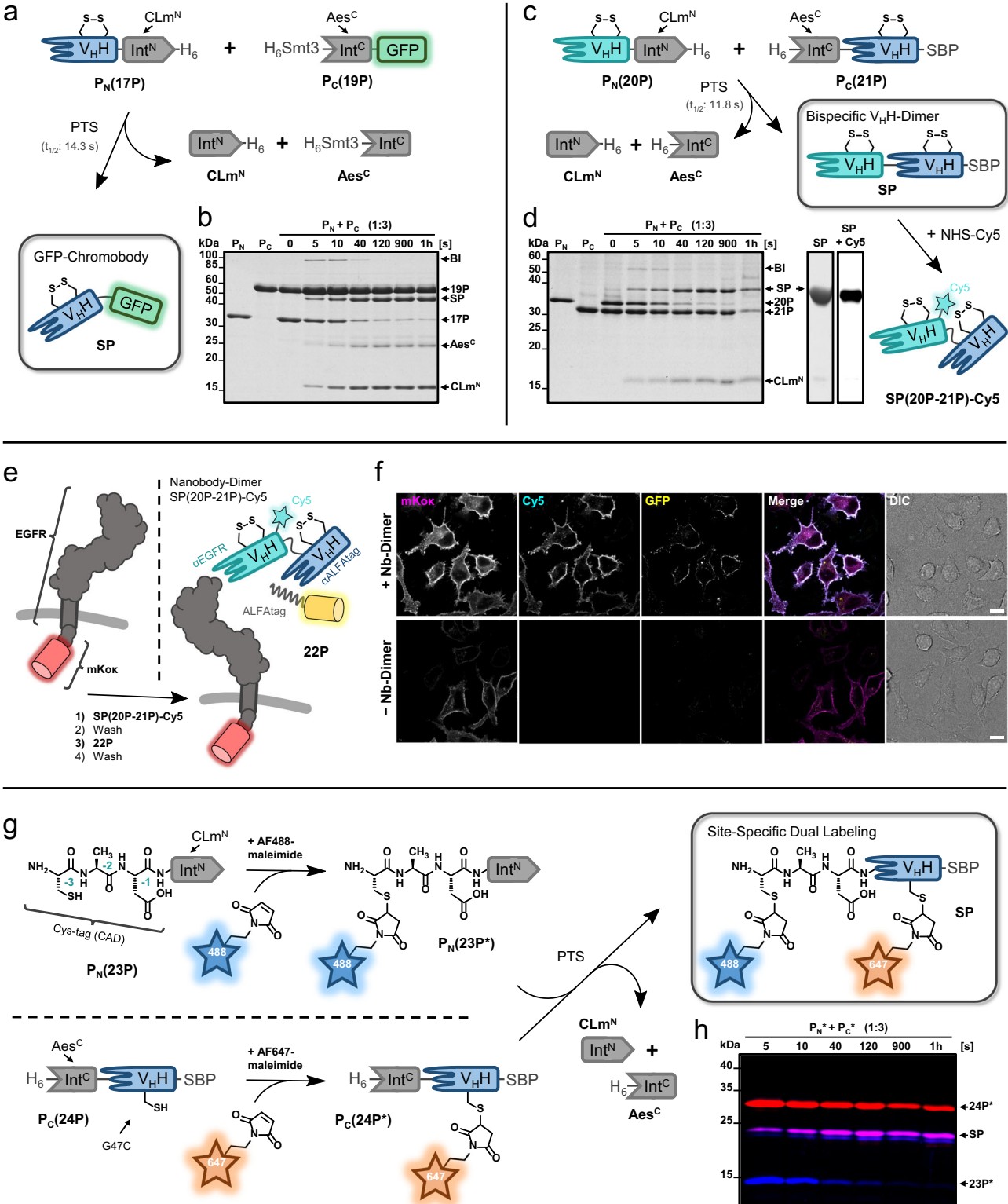

fragment (Supplementary Fig. 17b). Therefore, using the CLm$^N$ fragment for the intein precursor increased the yield of the isolated monomeric species by 13-fold. Importantly, this construct was also fully active in PTS, as demonstrated by a ligation with H$_6$-Smt3-Aes$^C$-GFP (**19P**) to furnish the ALFAnb-GFP chromobody as a splice product in nearly quantitative yields and with ultra-fast kinetics of t$_{1/2}$ = 14.3 s (Fig. 7a, b; Supplementary Fig. 17). Thus, the favorable properties of the CLm$^N$ precursor fragment were found to be general also in other extein contexts.

## CLm split intein-mediated protein modifications that highlight its cysteine-less nature

To further demonstrate the utility of the engineered CLm mutant we aimed for the ligation of two disulfide-containing proteins (Fig. 7). Biparatopic nanobody dimers are useful reagents to dimerize proteins but their expression can be problematic and less efficient compared to single nanobodies. To generate a dimer of the anti-EGFR nanobody EgA1 and the ALFA nanobody, we prepared, purified and incubated constructs EgA1nb-CLm$^N$-H$_6$ (**20P**) and H$_6$-Aes$^C$-ALFAnb-SBP (**21P**)

**Fig. 7 | Protein modification reactions using the engineered CLm$^N$ intein fragment. a** Scheme of the ALFAnb-GFP chromobody generation by PTS. **b** SDS-PAGE analysis of the PTS reaction shown in (**a**), at 37 °C, using the Aes$^C$ precursor at threefold molar excess (15 µM) (see also Supplementary Fig. 17). This experiment was repeated two times. **c** Scheme of the bispecific nanobody dimer generation by PTS and its fluorophore bioconjugation. Afterwards the generated V$_H$H-dimer **SP(20P-21P)** was purified and labeled with NHS-Cy5 to provide **SP(20P-21P)-Cy5**. **d** SDS-PAGE analysis of the PTS reaction shown in (**c**), at 37 °C, using the Aes$^C$ precursor at threefold molar excess (15 µM) (see also Supplementary Fig. 18). This experiment was repeated two times. The gel slices (right panel) show the purified splice product (SP) by Coomassie-staining and the Cy5-labeled SP with a fluorescence scan at 647 nm. **e** Scheme of the cell surface labeling by nanobody-dimer-mediated formation of a ternary complex. **f** Fluorescence-microscopy (CLSM) analysis of labeled cells (scale bar: 10 µm). **g** Scheme to prepare a nanobody using two separated thiol bioconjugation steps prior to PTS. **h** Analysis of formation of the dually labeled ALFAnb splice product (SP). Shown is a fluorescence scan of an SDS-PAGE analysis of the PTS reaction at 37 °C using the Aes$^C$ precursor **24P\*** in threefold molar excess (15 µM) (see also Supplementary Fig. 19). This experiment was repeated two times. Calculated molecular weights are: P$_N$(**17P**) = 30.7 kDa, P$_C$(**19P**) = 45.0 kDa, CLm$^N$-H$_6$ = 14.7 kDa, H$_6$-Smt3-Aes$^C$ = 18.4 kDa, SP(**17P-19P**) = 42.6 kDa, P$_N$(**20P**) = 30.7 kDa, P$_C$(**21P**) = 26.0 kDa, H$_6$-Aes$^C$ = 4.7 kDa, SP(**20P-21P**) = 36.4 kDa, P$_N$(**23P**) = 14.7 kDa, P$_N$(**24P**) = 26.8 kDa, CLm$^N$ = 13.7 kDa, SP(**23P\***−**24P\***) = 22.5 kDa. BI branched intermediate, SP splice product, V$_H$H variable domain heavy chain (nanobody).

(Fig. 7c, d). Again, highly efficient and rapid PTS occurred ($t_{1/2}$ = 11.8 s) (Supplementary Fig. 18a). The splice product EgA1nb-ALFAnb-SBP was purified, labeled with NHS-Cy5 and tested for biparatopic binding in a cellular assay (Fig. 7e, f). To this end, HeLa cells were transiently transfected to express the extracellular and *trans*-membrane domains of EGFR, while the intracellular kinase domain was displaced with the fluorescent protein mKoκ. Cy5-labeled EgA1nb-ALFAnb-SBP specifically bound only to transfected cells. We then added ALFA-tagged GFP (**22P**, GFP-ALFAtag-H$_6$) to the growth medium and observed the formation of the dimerized, three-color protein complex on the cell surface, dependent on the presence of the nanobody dimer, thus confirming its activity (Fig. 7f; Supplementary Fig. 18b). Notably, the utilized nanobody-containing split intein precursors were expressed in *E. coli* T7 shuffle cells to introduce their disulfide bonds and no reducing agents were added in the subsequent process to avoid their reduction. Both these operations would not have been suitable with cysteine-dependent split inteins.

To underline the orthogonality of the engineered CLm intein to thiol chemistries, we then aimed for site-specific dual protein labeling. We developed a new cysteine-tag[43] reagent for minimal chemical labeling based on the engineered CLm$^N$ fragment. The construct CAD-CLm$^N$ (**23P**) with a minimal N-extein composed of a three-residue cysteine-tag (Cys-Ala-Asp) was obtained by cleavage of a H$_6$-Smt3-CAD-CLm$^N$ protein with Ulp1 protease. We bioconjugated AF488-maleimide to the single N-terminal cysteine (Fig. 7g; Supplementary Fig. 19). Before transferring this labeled 3aa-tag to the protein of interest by PTS, this time using the ALFA nanobody as the C-extein, we introduced a surface-exposed[42] cysteine at position 47 to give H$_6$-CLm$^C$-ALFAnb(G47C)-SBP (**24P**). This single cysteine in the C-extein was then bioconjugated with AF647-maleimide. PTS with both labeled intein precursor proteins furnished the dually labeled ALFA nanobody virtually quantitatively and with ultra-fast splice kinetics ($t_{1/2}$ = 15.5) (Fig. 7h). Of note, the second singly labeled species visible below the splice product in Fig. 7h is an artifact from partial thioether cleavage during sample preparation (Supplementary Fig. 19). Again, this labeling scheme could only be realized with a cysteine-less split intein.

### Int$^N$ precursor aggregation is widespread in split inteins
We then applied the AMYLPRED2 prediction algorithm to other intein sequences. Interestingly, high prediction scores for amyloidogenic patterns were returned for many of the commonly used split inteins (Fig. 8a, b; Supplementary Fig. 20). Artificially split inteins even exhibited on average a higher score for such patterns in the second segment of their Int$^N$ fragment (N2), compared to naturally split inteins (Fig. 8c).

We selected the artificially split *Ssp* DnaB intein because this intein was previously described to consume its Int$^N$ precursor to only about 40% in the PTS reaction[29,43]. Using the model constructs MBP-Ssp$^N$-H$_6$ (**25P**) and MBP-Ssp$^C$-H$_6$ (**26P**)[29] we could reproduce the incomplete splicing of the Int$^N$ precursor (ca. 38%) (Fig. 8d, e). Indeed, analysis of **25P** by SEC revealed an inactive, high-molecular weight aggregate fraction (ca. 66%) next to a monomeric fraction (ca. 34%) (Fig. 8f). A larger quantity of the monomeric sample, prepared by our denaturation-refolding protocol, led to virtually complete splicing of the Int$^N$ precursor (Fig. 8d, e). To the best of our knowledge, this is the first time that full activity was described for this split intein.

As a second case, we investigated the naturally split *Npu* DnaE intein, known for its rapid and efficient splicing. However, certain precursor constructs have been reported to exhibit poor splicing yields of unknown origin[21]. We revisited the constructs GFP-Npu$^N$-H$_6$ (**27P**) and Npu$^C$-GFP (**28P**)[21], which yielded only about 45% splice product formation, again due to an inactive fraction of the Int$^N$ precursor **27P** (Supplementary Fig. 21). Indeed, SEC analysis revealed that this precursor existed to only 43% as a monomer, while the remaining fraction behaved as a high-molecular weight soluble aggregate. Using an isolated monomeric species of **27P** the splicing efficiency with **28P** increased to over 80%. Notably, the partially inactive **27P** was isolated at significantly higher concentrations than other Npu$^N$ precursors, suggesting that in this case the concentration dependency of the aggregation caused the splicing inefficiency.

## Discussion
In this work, we addressed the poor splicing efficiency of the Int$^N$ precursor of the cysteine-less Aes split intein, which is representative of a widespread and unexplained phenomenon observed for many split inteins. We carried out an in-depth biochemical and biophysical characterization of the intein combined with a structural and computational analysis. We revealed that the Aes$^N$ fragment is prone to form soluble high-molecular weight aggregates with high β-sheet content that are inactive when combined with an Aes$^C$ precursor, thus explaining its poor efficiency. By mutating residues in disordered regions that were computationally predicted for high propensity to serve as aggregation nuclei we could generate several triple and one quadruple mutant of the Aes$^N$ fragment that showed stable monomeric behavior and nearly complete splicing activity. Notably, the CLm$^N$ triple mutant (Cysteine-Less and monomeric) combined the high splicing efficiency with a retained ultra-fast splicing rate that places it among the fastest known split inteins, about 4-fold faster than the *Npu* DnaE intein[21]. The CLm intein (CLm$^N$ + Aes$^C$) thus holds great potential for expanded protein modification applications as a cysteine-less and high-activity split intein.

Importantly, we further showed that the explanation for the incomplete splicing efficiency seems generalizable. We computationally predicted similar aggregation features in several other split inteins and verified soluble aggregates to cause incomplete splicing of Int$^N$ precursors of the artificially split *Ssp* DnaB and the naturally split *Npu* DnaE inteins.

How can the general tendency of split inteins to form soluble aggregates be explained? Previous studies and our present work have revealed that a disordered-to-ordered transition is required to access the conserved intein horseshoe structure with its intertwined Int$^N$ and Int$^C$ fragments[35,36]. The conserved charge segregation, reflected by high and opposing local charges in the Int$^N$ and Int$^C$ pieces, helps both

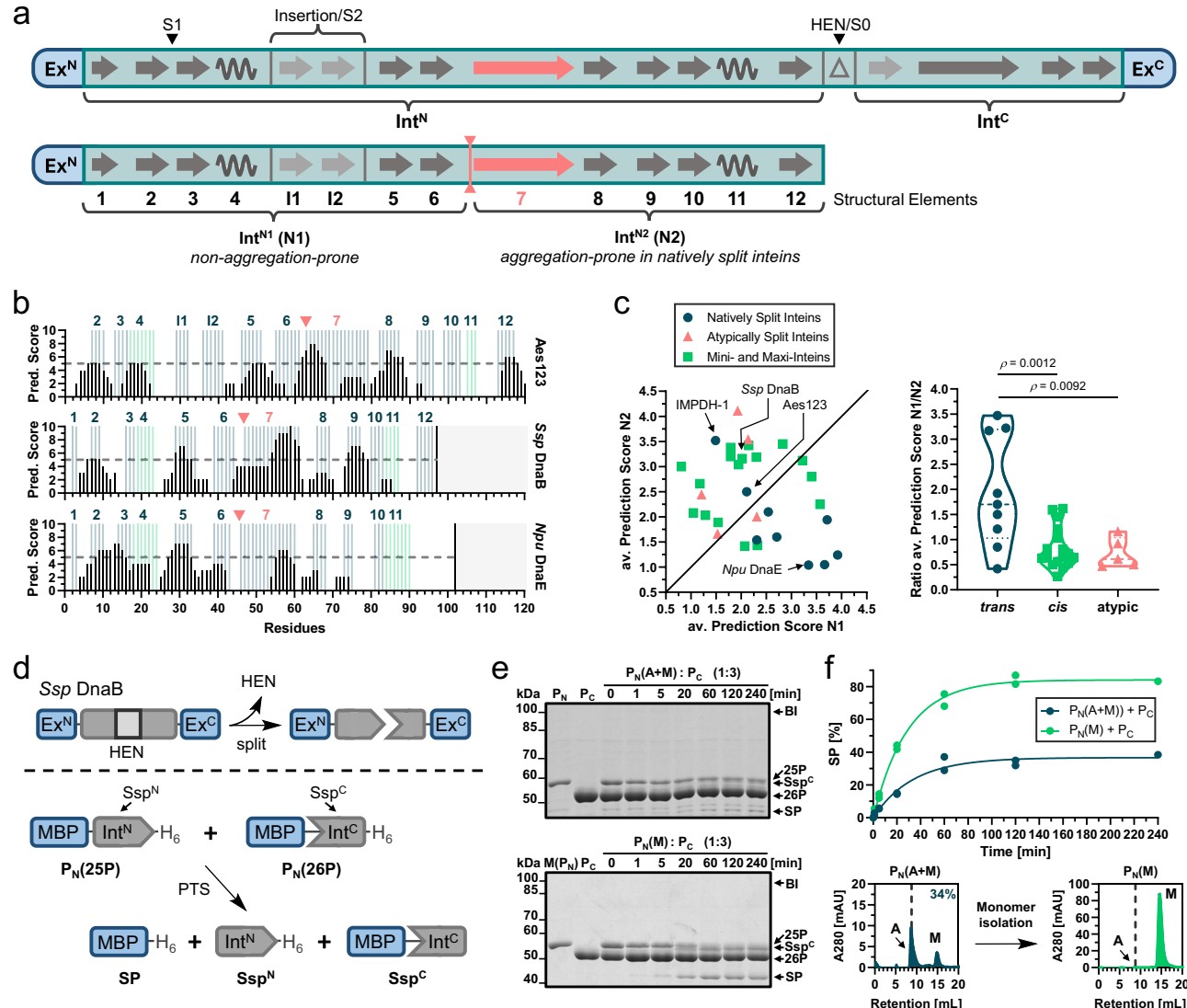

**Fig. 8 | Tendency to form inactive soluble aggregates is an inherent challenge for split inteins. a** Illustration of conserved secondary structure elements in the minimal intein horseshoe fold of *cis*-inteins (top) and the Int^N fragment of split inteins (bottom). Additionally, one optional insertion of β-strands I1 & I2 is shown that is also present in the Aes split intein. The split positions corresponding to typically split (HEN/S0) and atypically split (S1) inteins are indicated (HEN = homing endonuclease)[23,66]. The border between N1 and N2 segments was defined at the beginning of β-sheet 7 (vertical red line) in contrast to a symmetry axis within β-sheet 7 used previously[36]. **b** Consensus predictions of amyloidogenic pattern formation using AMYLPRED2 (ref. 40) of the Int^N parts of selected inteins. Secondary structure elements are marked in blue (β-sheets) and green (α-helices). **c** Plot of the average (av.) prediction scores of the Int^N(N2) segments against those of the Int^N(N1) segments of commonly used inteins (left panel). Distribution of the ratio of

the average scores (N1/N2) (right panel). **d** History of the artificially split *Ssp* DnaB intein (top)[29,67]. Scheme of the PTS reaction (bottom). **e** Coomassie-stained SDS-PAGE analysis of the PTS reactions as illustrated in **d** using a three-fold molar excess of **26P** (15 μM) at 37 °C. In the upper panel the *Ssp*^N precursor **25P** was used without prior SEC-purification, in the lower panel the isolated monomer of **25P** was used. These experiments were repeated two times. **f** Plot of the PTS reactions analyzed in (**e**) fitted to a one-phase exponential equation (top panel). SEC-analysis of **25P** before (left panel) and after (right panel) monomer isolation. Calculated molecular masses are: P_N(**25P**) = 56.4 kDa, P_C(**26P**) = 50.1 kDa, *Ssp*^N = 13.3 kDa, *Ssp*^C = 48.6 kDa, SP = 44.6 kDa. BI branched intermediate, SP splice product. For panel (**c**), n = 9 for natively split inteins, n = 17 for *cis*-inteins, n = 5 for atypically split inteins. *p*-values are derived from an ordinary one-way ANOVA Tukey´s multiple comparison test. Source data are provided as a Source Data file.

to keep the intein fragments in disordered conformations and to increase their association rate through electrostatic interactions. The disordered nature further enhances their association rate by enlarged capture radii. It also ensures the inactivity of the fragments prior to complex formation. In contrast to typical intrinsically disordered proteins[44], however, the disordered regions in split inteins have to exhibit a high content of β-sheet promoting residues (thus order-favoring residues) to finally stabilize the intein horseshoe fold. This propensity to form β-sheets creates an intrinsic conflict to balance out different possible folding pathways. While essential to adopt the stable horseshoe fold following association with the complementary Int^C

precursor, it also favors the formation of β-sheet-based homomeric aggregates.

In the cellular context, the association with the co-expressed, complementary Int^C precursor can kinetically outcompete the pathway towards aggregation. We therefore propose that the evolutionary pressure against the aggregation tendency has been insufficient for many split intein fragments. While it is challenging to predict the precise threshold at which aggregation tendency outweighs maintaining the disordered region in a folding-competent state, this notion explains the aggregation propensity of certain natural split intein precursors when produced in isolated form. It also accounts for the

dependence on protein concentration, as higher concentrations promote the nucleation and elongation processes of aggregation[45,46].

Importantly, these conclusions also align with the higher aggregation tendencies of artificially split inteins and the higher resistance to do so observed for atypically split inteins. Artificially split inteins are derived from *cis*-inteins in which the Int[N] is linked to the Int[C] fragment on the same polypeptide and hence was less exposed during evolution to form homomeric aggregates. A similar effect is obtained by the generation of caged split inteins through fusion of parts of their complementary sequence[47,48]. In atypically split inteins, on the other hand, the split site is shifted to give a short Int[N] fragment[23,24]. Thereby, the regions of opposing charges and disordered nature similarly become linked to a *cis*-arrangement in the large Int[C] fragment, hence reducing the aggregation propensity by intramolecular saturation. Consequently, atypically split inteins also show an altered association mechanism that relies more on hydrophobic interactions[49,50]. These considerations explain why artificially generated inteins with an atypical split site showed virtually complete splicing, in contrast to their parent constructs split at the endonuclease position, but lost their favorable association and splicing kinetics[12,29,34].

Finally, given that the molecular origins for the tendency of a split intein to form β-sheet aggregates are predictable from its own sequence, our rational design strategy, as demonstrated here for the Aes intein, is likely generalizable to other split inteins suffering from low splicing yields due to aggregation. Our work provides both the explanation and the cure for a folding flaw that has hampered split intein tools for more than 25 years.

## Methods

### Recombinant gene expression and protein purification

Plasmid-encoded constructs were expressed in *Escherichia coli* LOBSTR BL21 (DE3) Gold or Shuffle T7 cells grown in LB medium at 37 °C or 30 °C, respectively. Protein expression was induced at 28 °C for 4 h or 18 °C overnight by addition of either IPTG (0.4 mM) or L-arabinose (0.2% (w/v)), depending on the plasmid. Cells were ruptured using an Emulsiflex C5 (Avestin) or by sonication (10 s on/ 15 s off) for 12 min. The proteins **1 P** (and its mutated variants; see Supplementary Data 2 for cloning details), **4P, 6P, 11P–15P, 17P–20P, 22P, 25P–26P** fused to a hexahistidine tag were purified by Ni-NTA affinity chromatography at 4 °C using gravity flow columns (Cube Biotech) in Ni-NTA buffer (50 mM Tris, 300 mM NaCl, 20 mM imidazole, pH 8.0) and eluted with the same buffer containing 250 mM imidazole. Monomer isolation of **1P** and **4P** was achieved by cell rupture under denaturing conditions in Ni-NTA buffer with 8 M urea. The proteins were purified using a Ni-NTA gravity-flow column as described above and slowly refolded by stepwise dialysis into Ni-NTA buffer with 20% (w/v) sucrose as stabilizing osmolyte. The refolded protein was applied to size exclusion chromatography (SEC) performed on a Superdex 200 10/300 column (GE Healthcare) at 4 °C and a flow rate of 0.75 mL/min using a FPLC Äkta Purifier system (GE Healthcare). The eluted monomeric protein was directly used for subsequent assays.

The proteins **2P, 5P** and **28P** were expressed bearing an N-terminal H$_6$-Smt3 tag in *E. coli* LOBSTR BL21 (DE3) Gold cells grown in LB medium from an IPTG-inducible protein expression vector overnight at 18 °C. The cells were lysed by sonication and purified using a Ni-NTA gravity-flow column as described above. After further purification by SEC using a Superdex 200 10/300 column (GE Healthcare) at 4 °C and a flow rate of 0.75 mL/min the eluted protein was treated with 500 nM His$_6$-tagged Ulp1 for 30 min at 8 °C. After passing the protein mixture over Ni-NTA resin to remove Ulp1 and the cleaved Smt3 tag the desired product was obtained in the flowthrough.

**7P, 8P** and **23P** were generated with a DTD or CAD tripeptide N-extein, respectively. The H$_6$-Smt3-tagged proteins were expressed and purified as described above. In the case of purification under denaturing conditions (**7P, 8P**), the proteins were slowly refolded by stepwise dialysis in Ni-NTA buffer (50 mM Tris, 300 mM NaCl, pH 8.0) with 20% (w/v) sucrose. The refolded protein was applied to SEC performed on a Superdex 200 10/300 column (GE Healthcare) at 4 °C and a flow rate of 0.75 mL/min. The monomeric fractions were pooled and treated with 500 nM His$_6$-tagged Ulp1 and further purified as described above.

Proteins **3P, 21P, 24P** and **27P** bearing a streptavidin binding protein (SBP) tag were expressed in LOBSTR BL21 (DE3) Gold or Shuffle T7 cells (**24P**). Purification was achieved by a Strep-Tactin gravity-flow column (IBA) in buffer W (100 mM Tris, 150 mM NaCl, 1 mM EDTA, pH 8.0) and elution was induced by addition of buffer W with 2.5 mM desthiobiotin.

Tag-less protein purification of **9P** (and its mutated variants), **10P** and **16P** was achieved via chitin-binding domain (CBD) pulldown using the IMPACT™ kit (New England Biolabs) in CBD buffer (20 mM Tris, 500 mM NaCl, 1 mM EDTA, pH 8.0). The supernatant of the centrifuged cell lysate was transferred to a gravity flow column with chitin-agarose. On-column thiolysis was induced by adding 50 mM β-mercaptoethanol in CBD buffer to cleave of the fused *Ssp* GyrBN intein[51]. To achieve complete cleavage the column was left at 4 °C shaking for 48 h. Subsequently, the eluted protein was further purified by size exclusion chromatography (SEC) using Superdex 75 10/30 column (GE Healthcare) at 4 °C.

Protein concentrations were determined using the calculated extinction coefficient at 280 nm. The identity of the products was confirmed by ESI-MS and the purity was assessed by SDS-PAGE or analytical RP-HPLC (see Supplementary Data 1 for protein sequences).

### Protein *trans*-splicing assay

Reactions were started by mixing the N- and C-terminal intein precursor proteins in SP buffer (50 mM Tris, 300 mM NaCl, 1 mM EDTA, pH 7.0) at indicated concentrations at 37 °C and in the absence of reducing agents (unless otherwise stated). At indicated time points aliquots were removed and the reaction was stopped by adding 4x SDS-PAGE loading buffer (500 mM Tris/HCl, 8% (w/v) SDS, 40% (v/v) glycerine, 20% (v/v) β-mercaptoethanol, 5 mg/L bromophenol blue, pH 6.8) and boiling (95 °C, 5 min) or by adding 1% formic acid (final conc.). Splice product formation was analyzed by SDS-PAGE or ESI-MS, respectively.

### Preparation of bispecific nanobody by protein *trans*-splicing

Constructs **20P** and **21P** were purified as described, mixed in Ni-NTA buffer (20 mM Tris, 300 mM NaCl, pH 8.0) and incubated at 37 °C for 60 min. The reaction mixture was centrifuged (3000 × *g*, 3 min), loaded onto a Ni-NTA column and the nanobody dimer was collected in the flowthrough. Afterwards the nanobody dimer was enriched by strep-tactin affinity purification and eluted with buffer W + 2.5 mM desthiobiotin. The purified nanobody dimer was dialyzed against PBS buffer.

### Thiol-bioconjugation of CysTag proteins

The proteins **23P** (25 μM) and **24P** (35 μM) were purified as described above and reduced with 5 eq. TCEP for 15 min at 4 °C in PBS buffer (pH 7.4). Subsequently, AF488 and AF647 maleimide (Jena Bioscience), respectively, were added to each of the proteins in three steps starting with 2 eq. (60 min) followed by 1.5 eq. (30 min) twice at 4 °C. The reaction with **23P** was quenched by 8 eq. DTT and purified by a ZebaTM Spin Desalting Column (ThermoFisher). The labeled **24P** was further purified by Ni-NTA affinity chromatography.

### Densitometric analysis and determination of protein *trans*-splicing rate constants

Coomassie-stained bands were analyzed using Gel Analyzer (v2010a) and normalized to the corresponding molecular weight. Normalized

intensities were used to calculate the ratio $x$ of splice product to the precursor protein used in limiting amount to determine the splice yield as follows.

$$P(\%) = \frac{(100 \times x)}{100 + x} \qquad (1)$$

To determine the overall splice rate ($k_{total}$) the splice product formation was treated as a pseudo-first-order reaction with one of the precursors given in three-fold molar excess and plotted against time. The following singe exponential function was fitted to the data using GraphPad Prism (v8).

$$[P]_t = P_{\max}\left(1 - e^{-k_{total}t}\right) \qquad (2)$$

where $P$ is the normalized intensity of the splice product and $k_{total}$ describes the pseudo-first-order rate equation of the protein trans-splice reaction. The variable $t$ is the reaction time in seconds and $P_{max}$ is a normalization factor which represents the fraction of active precursor protein.

In order to fit a simplified three-state kinetic model as described by Shah et al.[52] to the experimental data, the normalized intensities of the precursor protein which was used in limiting amount [A], the normalized intensity of the branched intermediate [BI] and the normalized intensity of the splice product [P] were used. The global fit was based on a system of equations which are the analytical solution to the coupled differential rate equations for those species and carried out using GraphPad Prism (v8).

$$p = k_1 + k_2 + k_3$$

$$q = \sqrt{p^2 - 4(k_1 k_3)}$$

$$a = \frac{1}{2}(p + q)$$

$$b = \frac{1}{2}(p - q) \qquad (3)$$

$$[A]_t = P_{\max}\left[\left(\frac{k_1(a - k_3)}{a(a - b)}\right)e^{-at} + \left(\frac{k_1(k_3 - b)}{b(a - b)}\right)e^{-bt}\right]$$

$$[BI]_t = P_{\max}\left[\left(\frac{-k_1 a}{a(a - b)}\right)e^{-at} + \left(\frac{k_1 b}{b(a - b)}\right)e^{-bt}\right]$$

$$[P]_t = P_{\max}\left[\left(\frac{k_1 k_3}{ab}\right) + \left(\frac{k_1 k_3}{a(a - b)}\right)e^{-at} - \left(\frac{k_1 k_3}{b(a - b)}\right)e^{-bt}\right]$$

In these equations, $p$, $q$, $a$, and $b$ are algebraic combinations of rate constants $k_1$, $k_2$, and $k_3$. $P_{max}$ is analogous to the normalization factor described above.

## Analytical RP-HPLC and ESI-MS

RP-HPLC analysis was carried out using an Agilent 1260 Infinity series system (Agilent Technologies) with a multiple wavelength detector SL and a single quadrupole mass spectrometer (Agilent). Samples were diluted with 95% $H_2O$, 5% acetonitrile and 0.1% TFA and centrifuged (20,000 × $g$, 2 min). According to the sample concentration, an appropriate volume was loaded on an analytical C18 column (ZORBAX SB-C18 RR HT, 3 × 50 mm, 1.8 μm, Agilent) at a flow rate of 0.4 mL/min. After a desalting step for 3 min in 5% buffer B (eluent A: 0.1% formic acid in water; eluent B: 0.1% formic acid in acetonitrile) proteins were separated by gradual elution with 20–80% B in 11 min. Absorbance was recorded at 280 nm and subsequently the column was washed for 4 min with 100% B.

Mass analysis of intact proteins was performed using an Ulti-Mate™ 3000 RS system (Thermo Fisher Scientific GmbH) connected to a maXis II UHR-qTOF mass spectrometer (Bruker Daltonik GmbH) with a standard ESI source (Apollo, Bruker Daltonik GmbH). When necessary, proteins were reduced with 2 mM TCEP at 4 °C for 10 min to avoid inhomogeneity issues. Then, samples were acidified using a 10% formic acid solution to reach a pH 2-3 and centrifuged (20,000 × $g$, 3 min). According to the protein concentration, an appropriate volume of the supernatant was loaded on a C4 column (Advance Bio RP-mAb C4, 2.1 mm × 50 mm, 3.5 μm, Agilent Technologies) at a flow rate of 0.6 mL/min in 5% eluent B (eluent A: 0.1% formic acid in water; eluent B: 0.1% formic acid in acetonitrile). After a desalting period of 7 min at 5% B, a steep gradient was applied (5–60% B in 2 min). MS settings: capillary voltage 4500 V, endplate offset 500 V, nebulizer 5.0 bar, dry gas 9.0 L/min, dry $T$ = 200 °C, mass range $m/z$ 300–3000. Data were analyzed with DataAnalysis (v4.4) (Bruker Daltonik GmbH) and deconvolution was performed using the MaxEnt algorithm implemented in the software.

## Analytical size exclusion chromatography

Analytical gel filtration was done using a 1260 infinity LC system (Agilent) equipped with an AdvanceBio SEC 120 Å 1.9 μm, 2.1 × 150 mm PEEK (Agilent), AdvanceBio SEC 200 Å 1.9 μm, 2.1 × 150 mm PEEK (Agilent) or AdvanceBio SEC 200 Å 1.9 μm, 4.6 × 300 mm (Agilent) column at flowrates of 0.1 and 0.35 mL/min, respectively. Prior analysis, the proteins were diluted to 10 μM and incubated at 15 °C for 24 h in assay buffer (50 mM Tris, 300 mM NaCl, pH 7) to prevent the measurement of artifacts due to the concentration dependence and dynamics in aggregation. Then, the solution was directly measured at 15 °C by injecting 5 μL of the sample. The obtained UV profiles at 280 nm were normalized to the highest peak to determine the aggregate/monomer ratio by the integrated peak area using GraphPad Prism (v8) or Origin (v2024).

For the time-resolved measurements to monitor the kinetics of dis- and re-aggregation, the samples were directly injected from the same probe after concentration adjustment or monomer isolation (as described above) at the indicated time-points. A hyperbolic function was fitted to the aggregate or monomer ratio using GraphPad Prism (v8). Note that the start of the re-aggregation process was better fitted by a sigmoidal function. Proteins **17P**, **18P**, **25P** and **27P** were measured without prior concentration adjustment. Here, the cysteine-containing proteins were reduced with 0.5 mM TCEP prior SEC analysis to prevent dimer formation.

## Stokes radius determination

The Stokes radius of **1P** was determined by size exclusion chromatography on a Superdex 200 10/300 pre-packed column (GE Healthcare) at 4 °C and a flow rate of 0.75 mL/min in assay buffer. The elution volume of **1P** ($V_e$) was converted into the mobility-factor parameter ($K_{av}$) using the following equation.

$$K_{av} = \frac{V_e - V_0}{V_t - V_0} \qquad (4)$$

where $V_0$ is the column void volume determined with blue dextran and $V_t$ is the total column bed volume determined with acetone at 280 nm. The Stokes radius ($R_{ST}$) of **1P** was estimated using a linear calibration plot of $R_{ST}$ vs. $(-\log K_{av})^{1/2}$, obtained with the standard globular molecular weight markers β-amylase ($R_{ST}$ = 54 Å), alcohol dehydrogenase ($R_{ST}$ = 46 Å), bovine serum albumin ($R_{ST}$ = 35 Å), carbonic anhydrase ($R_{ST}$ = 21 Å) and cytochrome c ($R_{ST}$ = 17 Å) according to the method described elsewhere[53].

## Bioinformatic analysis

The charge hydrophobicity plot was done using the PONDR® software (http://www.pondr.com/)[54]. The mean hydrophobicity was determined using the Kyte-Doolittle hydrophobicity scale. In the plot, intrinsically disordered and native proteins are separated by a solid line which represents an empirically defined border as $R = 2.785\,H - 1.151$, where $R$ describes the mean net charge and $H$ the hydrophobicity[55]. The amino acid composition of Aes$^N$ was analyzed using ProtParam[56] and compared to the Disprot (v9.6)[44] and UniProtKB/TrEMBL (v2024_04) protein database[57].

For the prediction of aggregate-forming regions the sequence-based web tool AMYLPRED2[40] was used which employs a consensus of different methods to predict amyloid fibril formation. Note that the method AmyloidMutants usually integrated in the web tool was not used due to connection errors.

Protein structure prediction was done using Phyre2 (ref. 58) and AlphaFold3 (ref. 59).

## Biolayer interferometry (BLI)

Binding kinetics were measured using the Octet R8 instrument (Sartorius) and streptavidin conjugated biosensors (Sartorius) controlled by the build-in software Octet 21 CFR Part 11 (Sartorius). Sensor hydration was done in the final assay buffer (PBS, 0.02% Tween-20) for 10 min. Purified **5P** was biotinylated with 20 eq. biotin-X-N-hydroxy succinimide (Calbiochem) for 2 h at 4 °C. **6P** was diluted in assay puffer to final concentrations of 200 nM, 100 nM, 50 nM, 25 nM, 12.5 nM, 6.25 nM and 3.125 nM. The BLI assay was performed by: (1) Sensor equilibration: sensors immersed in assay buffer for 60 s. (2) Ligand loading: sensor immobilization with 2 μg/mL biotinylated **5P** for 300 s. (3) Baseline: sensors immersed in assay buffer for 120 s. (4) Association: sensors immersed with the analyte **6P** at different concentrations respectively for 500 s. (5) Dissociation: sensors immersed in assay buffer for 500 s.

## BLI data analysis

Due to the biphasic behavior in the binding response, the BLI data were analyzed based on analytical solutions of linear rate equations as described by Tiwari et al.[60]. BLI data fitting was achieved using double exponential functions according to the two-step conformational change model combining a bimolecular and a unimolecular equilibriums reaction using GraphPad Prism (v8) (Table 1). In general, the BLI response ($Y$) is a linear combination of the two variables

$$Y = \alpha X_1 + \beta X_2 \tag{5}$$

where $X_1$ is [N·C] and $X_2$ is [NC]. Here, [N·C] describes the associated complex and [NC] describes the folded complex. The association profiles were fitted using the equation:

$$Y = D + Ee^{-\sigma_1 x} + Fe^{-\sigma_2 x} \tag{6}$$

$$D = -(E + F)$$

where $D$, $E$ and $F$ are all constants. The exponents $\sigma_1$ and $\sigma_2$ are eigenvalues and dependent on the underling matrix (see Tiwari et al.[60]). The dissociation profiles were fitted using the following equation with two additional parameters:

$$Y = Ee^{-\gamma_1(x-t_0)} + Fe^{-\gamma_2(x-t_0)} \tag{7}$$

where $t_O$ is the time at the start of the dissociation phase. Since the $\alpha$ to $\beta$ ratio is unknown, both $E$ and $F$ are free fitting parameters. The exponents $\gamma_1$ and $\gamma_2$ can be obtained from the dissociation profile while $\gamma$ is $\sigma$ at the analyte concentration ($C$) $C = 0$. The dependencies of the individual fitted eigenvalues and the sums and products of these eigenvalues on the concentration $C$ were used to verify the correct binding model as described by Tiwari et al.[60].

To determine the rate constants, the slope of $\sigma_1 + \sigma_2$ plotted against $C$ was used to get $k_a$. The slope of $\sigma_1\sigma_2/k_a$ plotted against C gives the sum $k_f + k_u$, $\gamma_1 + \gamma_2 - (k_f + k_u)$ gives $k_d$, $\gamma_1\gamma_2/k_d$ gives $k_u$, and finally, $\gamma_1 + \gamma_2 - (k_d + k_u)$ provides $k_f$. Here, $k_a$ is the concentration dependent association rate, $k_d$ is the dissociation rate, $k_f$ describes the folding rate and $k_u$ describes the rate of unfolding. The equilibrium association constants were calculated as described below to get the overall association constant ($K_a$). Finally, the dissociation constant ($K_d$) was calculated as the inverse of $K_a$.

$$K_d = \frac{1}{K_a} \tag{8}$$

$$K_a = K_{a1}(1 + K_{a2})$$

$$K_{a1} = \frac{k_a}{k_d}$$

$$K_{a2} = \frac{k_f}{k_u}$$

The overall association and dissociation rate were calculated as described below:

$$k_{on} = \frac{k_a k_f}{k_d + k_f} \tag{9}$$

$$k_{off} = \frac{k_d k_u}{k_d + k_f}$$

For the steady state analysis, the average response ($Y$) from 490 to 500 s was used and the dissociation constant ($K_d$) was extracted from the following equation:

$$Y = \frac{Y_{\max} X}{(K_d + X)} \tag{10}$$

## Kinetic simulation

The two-step intein assembly mechanism consisting of a combined bimolecular equilibrium reaction and unimolecular equilibrium reaction was simulated by a numerical solution of the underlying rate equations (Table 1) using the built-in function ´NDSolve´ of Wolfram

---

**Table 1 | Reaction and rate equations of the biphasic two-step conformational change model**

| Reaction | Rate equation |
|---|---|
| $[N] + [C] \underset{k_d}{\overset{k_a}{\rightleftharpoons}} [N \cdot C] \underset{k_u}{\overset{k_f}{\rightleftharpoons}} [NC]$ <br> $[C] = [C_0] - [N \cdot C] - [NC]$ | $\frac{d[N \cdot C]}{dt} = k_a C[C_0] - (k_a C - k_d - k_f)[N \cdot C] - (k_a C - k_u)[NC]$ <br><br> $\frac{d[NC]}{dt} = k_f[N \cdot C] - k_u[NC]$ |

**Table 2 | Reaction and rate equations of the biphasic two-step conformational change model combined with a simplified one-step splice kinetic**

| Reaction | Rate equation |
|---|---|
| $[N]+[C] \underset{k_d}{\overset{k_a}{\rightleftharpoons}} [N \cdot C] \underset{k_u}{\overset{k_f}{\rightleftharpoons}} [NC] \overset{k_{splice}}{\rightarrow} [SP]$ | $\frac{d[N]}{dt} = -k_a C[N][C] + k_d[N \cdot C]$ |
| | $\frac{d[C]}{dt} = -k_a C[N][C] + k_d[N \cdot C]$ |
| | $\frac{d[N \cdot C]}{dt} = k_a C[N][C] - k_d[N \cdot C] - k_f[N \cdot C] + k_u[NC]$ |
| | $\frac{d[NC]}{dt} = k_f[N \cdot C] - k_u[NC] - k_{splice}[NC]$ |
| | $\frac{d[SP]}{dt} = k_{splice}[NC]$ |

Mathematica (v14.1). Here, [N] corresponds to the analyte concentration (C) and [C] describes the ligand concentration. $[C_O]$ is the initial ligand concentration bound to the sensor. [N·C] is the concentration of the associated complex and [NC] describes the concentration of the folded complex structure. The rate equations $k_a$, $k_d$, $k_f$ and $k_u$ are described in the BLI data analysis section (see above) and were determined experimentally.

The simplified three-step intein assembly and kinetic mechanism consisting of an additional and subsequent irreversible unimolecular reaction from [NC] to the splice product [SP] was similarly simulated by a numerical solution of the underlying rate equations (Table 2).

In the final simulation the rate determining step of folding was equated to the overall splice rate determined by Eqn. (2) ($k_f = k_{total}$) to account for the competition between $k_u$ and $k_{splice}$ which is not considered in the biolayer interferometry measurements due to construct inactivation. $k_3$ was determined as described by Eqn. (3) and equated with $k_{splice}$ as rate-determining step of the splice mechanism alone ($k_3 = k_{splice}$). To determine $k_3$ only the active precursor protein was considered to calculate the splice product formation.

## Circular dichroism spectroscopy

The isolated proteins **9P** (40 µM) and **8P** in its monomeric (16 µM) or aggregated (23 µM) form were briefly dialyzed into CD buffer (7.5 mM $K_2HPO_4$, 2.5 mM $KH_2PO_4$, 100 mM $(NH_4)_2SO_4$, pH 7.4). Far-UV circular dichroism spectra were obtained with a Jasco J-810 apparatus equipped with a 150 W air-cooled xenon lamp using a quartz cuvette with a 0.1 mm path length at 25 °C from 190 to 250 nm in CD buffer. Spectra were recorded at the indicated concentrations and CD buffer solution was used as blank. The protein denaturation experiments were performed with **8P** by adding 4 M or 6 M urea to the protein sample and CD spectra were obtained from 205 to 250 nm. The molar ellipticity was determined using the built-in spectrum analysis software.

## Thermal shift assay

Proteins **8P** and **9P** (and its mutated variants including **16P**) were adjusted to a final concentration of 0.2 g/L and 0.5 g/L, respectively, mixed with 5x SYPRO™ Orange (Invitrogen) as fluorescent dye and added to a 96-well PCR plate. The PCR plate was sealed and placed into CFX96 Touch Real-Time PCR Detection System (Bio-Rad) controlled by CFX Manager Software (v3.1) (Bio-Rad). Samples were heated from 10 to 90 °C in increments of 1 °C while fluorescence was measured. The measurements for each protein were performed 5-8 times. The data profile was cut at the highest value to account for post-peak aggregation of protein-dye complexes leading to quenching of the fluorescence signal. A sigmoidal four-parameter logistic equation was fitted to the truncated and normalized fluorescence imaging data using GraphPad Prism (v8).

$$Y = Y_{min} + \frac{Y_{max} - Y_{min}}{1 + 10^{(T_M - X) \times b}} o \qquad (11)$$

where $Y_{min}$ is the minimal normalized fluorescence intensity, $Y_{max}$ is the maximal intensity, $T_M$ describes the melting temperature, and $b$ describes the Hill coefficient.

## Carbene footprinting

The photochemical probe sodium 4-(3-(trifluoromethyl)-3H-diazirin-3-yl)-benzoatearyldiazirine (**1**) was prepared by treating 4-(3-(trifluoromethyl)-3H-diazirin-3-yl)-benzoic acid (TCI) with sodium hydroxide as described by Manzi et al. [61]. Also, the photochemical labeling approach was based on Manzi et al. Here, 10 µM **10P** and **8P** in its monomeric form ($n = 5$) were mixed with 1 mM aryldiazirine in a total volume of 30 µL as drop placed on the lid of a microreaction tube in 50 mM Tris, 300 mM NaCl, pH 8. The mixture was left equilibrating for 5 min at RT in the dark. Then, the solution was snap-frozen in liquid $N_2$ (77 K) and placed on dry ice. The labeling reaction was initiated by 2 s of irradiation with 365 nm at 1400 mA using a LED lamp M365LPI (Thorlabs Inc.). The distance to the probe was kept constant at 5 cm. After irradiation, the sample was thawed at RT.

For tryptic protein in-gel digestions, the labeled mixture was then mixed with 4x SDS loading dye (250 mM Tris/HCl, pH 6.8, 8% (w/v) SDS, 40% (v/v) glycerine, 20% (v/v) β-mercaptoethanol, 0.2% (w/v) bromophenol blue) and heated to 95 °C for 5 min. Following separation by SDS-polyacrylamide gel electrophoresis and Coomassie brilliant blue staining, protein bands were excised, destained by 50% (v/v) EtOH in $H_2O$, 0.1% (v/v) TFA at 60 °C overnight, washed and dried with acetonitrile and by vacuum concentration. Trypsin digestion was performed at 37 °C using 400 ng pre-warmed trypsin (Promega) in 50 mM ammonium carbonate supplemented with 0.01% ProteaseMax (Promega) for 6 h. Afterwards, the supernatant (20 µL) was collected and acidified with 0.1% formic acid as final concentration and analyzed by LC-MS without further dilution.

For the LC-MS analysis of the tryptic peptides an UltiMate™ 3000 RS system (Thermo Fisher Scientific GmbH) connected to a maXis II UHR-qTOF mass spectrometer (Bruker Daltonik GmbH) with a standard ESI source (Apollo, Bruker Daltonik GmbH) was used. An appropriate amount of the peptide solution was loaded on a C18 column (ZORBAX SB-C18 RR HT, 80 Å, 1.8 µm, 50 mm×3 mm, Agilent Technologies) at a flow rate of 0.6 mL/min in 5% eluent B (eluent A: 0.1% formic acid in water; eluent B: 0.1% formic acid in acetonitrile). After 5 min at 5% B, a gradient was applied (5 - 35% B in 20 min, followed by 35 - 100% B in 2 min, 1 min at 100% B and 2 min at 5% B). The following MS settings were applied: Positive polarity, capillary voltage 4500 V, endplate offset -500 V, nebulizer 1.5 bar, dry gas 8.0 L/min, dry T = 180 °C, mass range m/z 150-2200.

To analyze the data of the carbene-labeled peptides, MSconvert GUI (64-bit, v3.0.23046-c1e4e67) was used to convert Bruker file folders containing .baf files to mz5 files. Settings: Binary encoding precision 64-bit, Write index, Use zlib compression, TPP compatibility were ticked. Filter: "Zero samples" with parameters: "removeExtra 1-". The mz5 files were used to identify labeled and unlabeled peptides and to determine the fractional modifications by means of the PepFoot

software (v1.2) as described elsewhere[62]. PepFoot settings: Aryldiazirin-TDBA modification of trypsin-digested peptides with a length of 5-20 amino acids and charge states of 1-8. Tolerance: 15 mmu. Miss-cleavage of peptides was not considered.

Tandem MS analysis of tryptic peptides (LC-MS2) was performed using an UltiMate™ 3000 RS LC nano system (Thermo Fisher Scientific GmbH) connected to a maXis II UHR-qTOF mass spectrometer with a nano-ESI source (CaptiveSpray with nanoBooster, Bruker Daltonik GmbH). 3.5 μL of each sample were loaded on a C18 trapping column (Acclaim PepMap™ 100, 5 μm, 100 Å, ID 100 μm x L 20 mm, Thermo Fisher Scientific GmbH) at a flow rate of 20 μL/min in 2% eluent B (eluent A: 0.1% formic acid in water; eluent B: 0.1% formic acid in acetonitrile). After 10 min of washing at 2% B, a 45-minute gradient (5 to 50% B, flow rate 500 nL/min) was applied for the separation on a C18 nano column (PepSep TWENTY-FIVE C18, ID 150 μm x L 250 mm, 1,5 μm, Bruker Daltonik GmbH). MS settings: capillary voltage 1600 V, mass range: m/z 150-2200. MS survey scans were performed with a cycle time of 2.5 s. After each survey scan, the 10 to 20 most abundant precursor ions with z > 1 were selected for fragmentation using collision-induced dissociation. MS/MS summation time was adjusted depending on the precursor intensity, the precursor isolation window and the collision energy were depending on the precursor m/z and charge. DataAnalysis (v5.3) (Bruker Daltonik GmbH) was used for chromatogram processing and fragment spectra isolation. The resulting mgf files were analyzed using ProteinScape (v4.2) (Bruker Daltonik GmbH) as a front-end for searches against the SWISS-PROT databases on a Mascot server (Mascot 2.5, Matrix Science Ltd.; Supplementary Data 3).

### Protein crystallization and structure determination

Purified **9P** was crystallized using the hanging drop method and microseeds streak seeding with a cat whisker. Best crystals were grown at 20 °C with a protein concentration of 7 mg/ml in 50% precipitant mix (40% v/v PEG 500* MME; 20% w/v PEG 20000), 0.1 M buffer system (1 M Tris base, BICINE) pH 8.5, and 0.1 M monosaccharides as additive. X-ray diffraction data were collected at beamline P13 (EBML, DESY, Hamburg) to 1.38 Å resolution (Supplementary Table 4). Molecular replacement was done with a Phyre2 model (ref. [58]) using the Aes123 PolB1 sequence and an initial model was obtained using Phenix AutoSol[63]. The structure was further built with Coot[64] and refined with Refmac5 (ref. [65]). Four terminal N-extein residues were resolved in the electron density as well as seven residues of the C-extein. The coordinates and structure factors have been deposited in the Protein Data Bank under the accession codes 9HTH.

### Cell culture

HeLa cells (Merck, CB_93021013) were cultured in EMEM (supplemented with 10% fetal calf serum, 1% non-essential amino acids and 1% L-glutamine) at 37 °C and 5% $CO_2$. $1.8 \times 10^5$ confluent cells were seeded on 16 mm coverslips in 12-well plates and used for transient transfection with the respective pDisplay plasmid, encoding EGFR-mKoκ using Metafectene (Biontex) as recommended by the manufacturer. After 36 h of incubation, binding studies and microscopy analysis were performed. To this end, cells were incubated with 250 nM of Cy5-labeled EgA1-ALFA nanobody-dimer in fresh medium for 10 min at 37 °C. Afterwards, cells were incubated with 50 nM of GFP-ALFAtag-$H_6$ (**22P**) for another 5 min, washed twice with PBS prior to fixation. Cy5-labeling of the nanobody dimer was achieved with 20 eq. biotin-X-N-hydroxy succinimide (Calbiochem) for 2 h at 4 °C. The reaction was quenched with 10 mM Tris and the labeled protein was purified by Strep-Tactin affinity chromatography before applied to the cells.

### Confocal laser scanning microscopy

Cells were fixed with 4% paraformaldehyde in PBS for 20 min at 25 °C, washed with $H_2O$ and mounted on coverslips using Aqua/Poly-Mount mounting solution (Polysciences). Confocal microscopy was carried out on a Leica TCS SP8 system using a Leica DMi8 inverted microscope equipped with a HC PL APO 63x/1.20 W CORR UVIS CS2 water-immersion objective lens. Signal detection was done with a hybrid detector (HyD™) in photon counting mode using the Leica acquisition software LAS X (v3.5.6.21594). Image processing was done with imageJ (v2.14). An image format of 1024 px² was used for sample acquisition, giving a final pixel size of 180 nm.

### Statistics and reproducibility

No statistical method was used to predetermine the sample size. However, we conducted relevant experiments with sample sizes sufficient to determine statistically significant differences across samples. No data were excluded from the analyses. The experiments were not randomized, and investigators were not blinded to allocation during experiments or outcome assessment. All statistical tests on experimental groups were performed using GraphPad Prism software (v8).

### Reporting summary

Further information on research design is available in the Nature Portfolio Reporting Summary linked to this article.

## Data availability

The quantitative LC-MS carbene footprinting data generated in this study have been deposited in the datastore database under accession code https://doi.org/10.17879/73998391189. The data are available under CC BY-NC 4.0 license. The processed LC-MS2 data are available in Supplementary Data 3. Source data are provided as Source Data file. Source data are provided with this paper. The crystallographic data generated in this study have been deposited in the PDB database under accession code 9HTH https://doi.org/10.2210/pdb9hth/pdb. Furthermore, we used in this study crystallographic data with accession code 8CPN https://doi.org/10.2210/pdb8CPN/pdb[13]. Expression plasmids pCH145 and pCH196 encoding CLm$^N$ and Aes$^C$ split intein precursors **15P** and **2P**, respectively, have been deposited at Addgene (Addgene ID: 234487 and 234486). Source data are provided with this paper.

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

## Acknowledgements

This project was financially supported by the Deutsche Forschungsgemeinschaft (grants DFG MO1073/6-2 and MO1073/9–1 to H.D.M.). We thank Dr. Jedd Bellamy-Carter (Loughborough University) for kindly providing his PepFoot software.

## Author contributions

C.H. conceptualized the study, performed all experiments except for protein crystal structure determination, analyzed data and wrote the manuscript. Z.Y. performed DNA cloning and protein preparation for crystallography. K.F. performed crystallographic data collection and refinement. W.D. methodologically established carbene footprinting and supported data analysis. D.K. assisted in crystallographic data collection and refinement and reviewed the manuscript. H.D.M. conceptualized the study, supervised experiments and wrote the manuscript. All authors read and approved the final manuscript.

## Funding

## Competing interests

There are no competing interests.
