## [Transparent Peer Review file · Nature Communications]

A Cysteine-Less and Ultra-Fast Split Intein Rationally Engineered from Being Aggregation-Prone to Highly Efficient in Protein *trans*-Splicing

Corresponding Author: Professor Henning Mootz

Version 0:

Reviewer comments:

Reviewer #1

(Remarks to the Author)

Split inteins are extremely useful tools that have led to the development of several techniques in protein engineering. In particular, the report of the ultrafast, cysteine-less Aes split intein (Bhagawati et al. PNAS 2019) was a significant discovery. Unfortunately, the Aes intein was hampered by relatively low splicing efficiency. The manuscript from Humberg et al. details the development of a modified version of the Aes intein, termed CLm, with dramatically improved splicing efficiency while maintaining ultrafast kinetics.

The authors first demonstrate that the Aes N-intein (AesN) forms both monomers and soluble aggregates using size exclusion chromatography, with on the monomers remaining splicing competent. These aggregates increase in a concentration-dependent manner, limiting the use of the Aes split intein system. The authors next show that while the AesN monomer is largely unstructured, the AesN soluble aggregates adopt a secondary structure rich in β -sheets. Using chemically labeling, mass spec, and size exclusion chromatography, the authors identify disordered regions responsible for aggregation. Next, the authors use a web tool used to aggregation prone regions and combined with X-ray crystallography, identify 16 mutations that are predicted to reduce the propensity of AesN to aggregate. As expected, these mutations increase splicing and reduce AesN aggregation, and the authors combine three amino acid substitutions to form the CLmN intein. CLmN has substantially improved properties compared to the AesN, with near complete splicing efficiency and the tendency to avoid soluble aggregates even at relatively high (120 μ M) protein concentrations. In addition to showing that CLmN can accommodate numerous proteins as the N-extein, the authors further demonstrate the novel utility of this intein by performing splicing reactions otherwise impossible with a cysteine-containing split inteins. Finally, the authors find that IntN polypeptides from other split inteins also tend to form unreactive soluble aggregate, providing a roadmap to improve the activity of other widely used split intein.

As a whole, this work represents a significant advance in the field and the manuscript should be accepted following minor revisions. The experimental data presented is strong and supports the interpretations and conclusions. That said, I strongly believe that the manuscript would benefit tremendously from more careful editing, as the writing is sloppy in places.

Reviewer #2

(Remarks to the Author)

The manuscript submitted by Humberg et al. for consideration in Nat. Commun. describes the fast but incomplete splicing activity of a split Ae123 PolB1 intein and how to overcome the incomplete splicing. The authors identified that β -sheet formation and aggregation at high concentrations was the cause of reduced splicing efficiency. Computational, biochemical, and biophysical studies of the monomeric fraction revealed sequence regions prone to aggregation and allowed the design and production of triple mutant CLm intein that prevents aggregation but retained the ultra-fast splicing rate discovered for the wild-type.

Overall, the manuscript is well written, the figures neatly presented and the methods described in good detail, with results supporting the authors conclusions. The authors have presented a thorough and detailed investigation of the cause of incomplete splicing – a challenge that is shared by several widely-used inteins – but so far has not been well understood.

The manuscript presents an extensive and well-executed body of work, with a clear rationalization and improvement in the splicing activity of the triple mutant Clm intein and potential impact for the chemical biology community due to its orthogonality to cysteine-containing inteins. Therefore, I recommend the publication of this article in Nat. Commun. In revising the manuscript, the quality of the paper might benefit from the considerations of the points listed below:

- The introduction provides a broad overview of inteins and demonstrates the authors' expertise on this topic, but appears to follow an inconsistent flow of the argument, which might be confusing for the reader and dilute the key information needed to introduce the work in this manuscript. In contrast, the results and discussion section gives an easy-to-follow and logical description of observations and interpretations. In revising the manuscript, the authors might consider shortening the introduction and focusing on the information critical for this work. Figure 1a is rather oversimplified and it might be helpful to have a dedicated conceptual introductory figure in which the level of detail and labels in the figure is balanced with the details and terms used in the text e.g., IntN, IntC, PN, PC, N1, and N2 lobes.

- The figures are generally well designed and professionally presented. In some cases, the labels and compound numbering were difficult to follow alongside the text e.g., AesN precursor, PN, 1P, and MBP-AesN-H6. In Figure 1a, the asparagine side-chain should be corrected, and it is not clear at which point the -His6 tag is cleaved off (if it is?).

- It is interesting to see (Figure 2a) that the AesN precursor appears to form only two species – the monomeric form, and a large (but soluble) aggregate equivalent to approximately 29 monomeric units. The authors could comment on this, or add any observation of smaller oligomeric species (dimers, trimers etc.) or larger insoluble aggregates for some of the constructs or mutants.

- As noted by the authors, the rational approach of mutating residues likely to be involved in aggregation is applicable to other inefficiently-splicing inteins and it would be helpful to label the positions of the mutated residues (e.g. in the most efficient triple mutant T69K/F75H/M118N) on the crystal structure e.g. in Figure 4c. In the discussion, the authors thoughtfully rationalized how the mutations could affect aggregation and splicing efficiency. A comment about how these insights into which mutations at which sites decreased aggregation could advise the engineering of other aggregation-prone low-efficiency inteins, would increase the potential impact of this important work.

Reviewer #3

(Remarks to the Author)

This exciting manuscript describes a relatively simple approach for curing aggregation in N-terminal intein segments. The authors are correct that this has been a major hinderance for a wide variety of intein splicing applications, and this work therefore has significant potential impacts in several fields.

The work is explained well, and the experimental results are compelling. Some of the language in the manuscript is a little confusing, but becomes clear when read carefully.

I only have a couple of minor corrections:

Line 42, the word, "been" should be deleted.

Line 71, is the word, "inteinβs" correct?

Reviewer #4

(Remarks to the Author)

This is an important and well written manuscript describing the advancement of split intein technology to improve its independence of redox conditions. The approach for intein optimization has high novelty. I think that this work will be of interest to a wide audience. I found the experiments persuasive and only have minor suggestions:

Good to say a little more on the origin and species of Aes.

If there are two species coming together, is the reaction 2nd order overall and wouldn't it be more informative to present 2nd order rate constants? Are these half-times that are presented only relevant at the particular concentration chosen?

Explain how C- and N-cleavage occur.

Make it easier to find the residue size of each part.

Comment on retained residues on each extein that help optimal reaction.

Normally the binding loops on VHH are close to the N-terminus. Just to confirm is that the correct orientation in Fig. 6 when the binding loops look like they are close to the C-terminus?

The methods are not clear on the "Protein trans-splicing assay" e.g. describe the buffer more directly.

"Subsequently, Alexa Fluor 488 and Alexa Fluor 647 maleimide (Jena Bioscience), respectively, were added to each of the proteins in three steps"

Usually these dyes are sold by Thermo Fisher and Jena have different dyes?

Stokes, not Stoke, radius

Fig. S1b: curly arrow at stage 3 not well positioned

Fig. S8a: include chirality

Fig. S9c/d/e: typos in axis labels

angel instead of angle at various places

Answers to the reviewers:

Reviewer #1:

Split inteins are extremely useful tools that have led to the development of several techniques in protein engineering. In particular, the report of the ultrafast, cysteine-less Aes split intein (Bhagawati et al. PNAS 2019) was a significant discovery. Unfortunately, the Aes intein was hampered by relatively low splicing efficiency. The manuscript from Humberg et al. details the development of a modified version of the Aes intein, termed CLm, with dramatically improved splicing efficiency while maintaining ultrafast kinetics.

The authors first demonstrate that the Aes N-intein (AesN) forms both monomers and soluble aggregates using size exclusion chromatography, with on the monomers remaining splicing competent. These aggregates increase in a concentration-dependent manner, limiting the use of the Aes split intein system. The authors next show that while the AesN monomer is largely unstructured, the AesN soluble aggregates adopt a secondary structure rich in β -sheets. Using chemically labeling, mass spec, and size exclusion chromatography, the authors identify disordered regions responsible for aggregation. Next, the authors use a web tool used to aggregation prone regions and combined with X-ray crystallography, identify 16 mutations that are predicted to reduce the propensity of AesN to aggregate. As expected, these mutations increase splicing and reduce AesN aggregation, and the authors combine three amino acid substitutions to form the CLmN intein. CLmN has substantially improved properties compared to the AesN, with near complete splicing efficiency and the tendency to avoid soluble aggregates even at relatively high (120 μ M) protein concentrations. In addition to showing that CLmN can accommodate numerous proteins as the N-extein, the authors further demonstrate the novel utility of this intein by performing splicing reactions otherwise impossible with a cysteine-containing split inteins. Finally, the authors find that IntN polypeptides from other split inteins also tend to form unreactive soluble aggregate, providing a roadmap to improve the activity of other widely used split intein.

As a whole, this work represents a significant advance in the field and the manuscript should be accepted following minor revisions. The experimental data presented is strong and supports the interpretations and conclusions. That said, I strongly believe that the manuscript would benefit tremendously from more careful editing, as the writing is sloppy in places.

We thank the reviewer for the positive evaluation. We have carefully read the manuscript and included several little corrections throughout the manuscript to improve the writing. In particular, we have largely rewritten large parts of the introduction (also in response to reviewer 2), which indeed required improvement at several places.

Reviewer #2 (Remarks to the Author):

The manuscript submitted by Humberg et al. for consideration in Nat. Commun. describes the fast but incomplete splicing activity of a split Ae123 PolB1 intein and how to overcome the incomplete splicing. The authors identified that β -sheet formation and aggregation at high concentrations was the cause of reduced splicing efficiency. Computational, biochemical, and biophysical studies of the monomeric fraction revealed sequence regions prone to aggregation and allowed the design and production of triple mutant CLm intein that prevents aggregation but retained the ultra-fast splicing rate discovered for the wild-type.

Overall, the manuscript is well written, the figures neatly presented and the methods described in good detail, with results supporting the authors conclusions. The authors have presented a thorough and detailed investigation of the cause of incomplete splicing – a challenge that is shared by several widely-used inteins – but so far has not been well understood. The manuscript presents an extensive and well-executed body of work, with a clear rationalization and improvement in the splicing activity of the triple mutant CIm intein and potential impact for the chemical biology community due to its orthogonality to cysteine-containing inteins. Therefore, I recommend the publication of this article in Nat. Commun.

We thank the reviewer for these positive words.

In revising the manuscript, the quality of the paper might benefit from the considerations of the points listed below:

- The introduction provides a broad overview of inteins and demonstrates the authors' expertise on this topic, but appears to follow an inconsistent flow of the argument, which might be confusing for the reader and dilute the key information needed to introduce the work in this manuscript. In contrast, the results and discussion section gives an easy-to-follow and logical description of observations and interpretations. In revising the manuscript, the authors might consider shortening the introduction and focusing on the information critical for this work.

Indeed, the introduction contained some inconsistencies in the flow of information. However, we believe the relatively large amount of information is necessary because the experimental work and discussion builds on so many aspects of intein research, for example applications of inteins, cis-inteins vs. split inteins, Cys-dependent vs. cysteine-less split inteins, advantages of cysteine-less split inteins, limitations of currently available cysteine-less split inteins, general limitations of split inteins (in particular with regard to splicing efficiency), different types of split inteins and their poorly understood incomplete splicing behavior, as well as folding states of split intein fragment prior and after assembly into the active intein complex.

We have rewritten the introduction in large parts in the revised manuscript to improve the flow of the arguments and removed some unnecessary information (shortening by about 7%).

Figure 1a is rather oversimplified and it might be helpful to have a dedicated conceptual introductory figure in which the level of detail and labels in the figure is balanced with the details and terms used in the text e.g., IntN, IntC, PN, PC, N1, and N2 lobes.

We thank the reviewer for this suggestion. We have now included an additional introductory figure (Figure 1 in the revised manuscript) to address these points and further support the reader with the key background information on split inteins.

The figures are generally well designed and professionally presented. In some cases, the labels and compound numbering were difficult to follow alongside the text e.g., AesN precursor, PN, 1P, and MBP-AesN-H6.

The terms MBP-AesN-H6 and 1P are always used together. We use the term AesN precursor or PN (as abbreviation) when statements become more general. The way we use these terms thus makes sense in the current form in our opinion.

In Figure 1a, the asparagine side-chain should be corrected, and it is not clear at which point the -His6 tag is cleaved off (if it is?).

Thank you, corrected. In the product, the side chain was also changed to the succinimide. The His6-tag is not cleaved off and is now shown.

- It is interesting to see (Figure 2a) that the AesN precursor appears to form only two species – the monomeric form, and a large (but soluble) aggregate equivalent to approximately 29 monomeric units. The authors could comment on this, or add any observation of smaller oligomeric species (dimers, trimers etc.) or larger insoluble aggregates for some of the constructs or mutants.

Please note that we cannot calculate the number of how many units the aggregated form contained as the protein eluted with the void volume. We never observed smaller aggregates (dimers, trimers). Even in the re-aggregation experiment, the aggregates directly appear as these high-molecular weight oligomers. We do not know the exact polymerisation process and did not investigate this further in detail. But we have included a short statement that no smaller oligomeric species of dimers or trimers were detected. “No smaller oligomeric species of dimers or trimers etc. were detected.”

- As noted by the authors, the rational approach of mutating residues likely to be involved in aggregation is applicable to other inefficiently-splicing inteins and it would be helpful to label the positions of the mutated residues (e.g. in the most efficient triple mutant T69K/F75H/M118N) on the crystal structure e.g. in Figure 4c.

We have included red arrows in Figure 4c to highlight these three residues.

In the discussion, the authors thoughtfully rationalized how the mutations could affect aggregation and splicing efficiency. A comment about how these insights into which mutations at which sites decreased aggregation could advise the engineering of other aggregation-prone low-efficiency inteins, would increase the potential impact of this important work.

We believe that our rationale to choose the mutations is sufficiently described in the results section. We describe that initially 10 positions were chosen in regions based on the computational aggregation prediction and the mutations at these positions were guided by the crystal structure, but manually picked, to avoid potential deleterious effects (mutations were selected to fit sterically, to lower the propensity to form β -sheets, and mutations towards polar and positively charged side chains were favored). We believe that likely many other mutations at these and other positions in the aggregation hot spots would have likewise resulted in significant improvements.

Reviewer #3:

This exciting manuscript describes a relatively simple approach for curing aggregation in N-terminal intein segments. The authors are correct that this has been a major hinderance for a wide variety of intein splicing applications, and this work therefore has significant potential impacts in several fields.

The work is explained well, and the experimental results are compelling. Some of the language in the manuscript is a little confusing, but becomes clear when read carefully.

We thank the reviewer for the positive comments. Please also see our responses to reviewers #1 and #2 with regard to changes in the text.

I only have a couple of minor corrections:

Line 42, the word, “been” should be deleted.

Corrected, thank you.

Line 71, is the word, “inteinβs” correct?

Thank you for spotting this typo. This sentence has been deleted in the revised version in our effort to rewrite and improve the introduction (see comments to reviewers #1 and #2).

Reviewer #4:

This is an important and well written manuscript describing the advancement of split intein technology to improve its independence of redox conditions. The approach for intein optimization has high novelty. I think that this work will be of interest to a wide audience. I found the experiments persuasive and only have minor suggestions:

Good to say a little more on the origin and species of Aes.

We have included the following statement upon first mentioning of the intein in the introduction: “found inserted in PolB-type DNA polymerase genes from T4-like bacteriophages of *Aeromonas salmonicida*”.

If there are two species coming together, is the reaction 2nd order overall and wouldn't it be more informative to present 2nd order rate constants? Are these half-times that are presented only relevant at the particular concentration chosen?

Yes, in principle these are bimolecular reactions. However, to simplify the analysis, we have determined the pseudo-first order rate constants by always adding one split intein precursor in 3-fold molar excess over the other to ensure that the concentration of the partner protein added in excess can be regarded as constant. This is the most common way in the literature to characterize split inteins.

Explain how C- and N-cleavage occur.

We have included a link to the Supplementary Figure 1 at the first mentioning of the side reactions.

Make it easier to find the residue size of each part.

We have moved this information from later in the results section up to the first sentence of the results section. “with its AesN (120 aa) and AesC(39 aa) fragments”. We have also included this information in the legend of the Figure 2 in the revised manuscript (formerly Figure 1).

Comment on retained residues on each extein that help optimal reaction.

On first mentioning of the experimental IntN and IntC precursor proteins in the first paragraph of the results section, we have included the sentence: “Three native flanking residues were kept on each side of the intein fragments here and throughout this work unless stated differently (DTD and SVY, respectively).”.

Normally the binding loops on VHH are close to the N-terminus. Just to confirm is that the correct orientation in Fig. 6 when the binding loops look like they are close to the C-terminus?

Thank you for this comment, we have changed the Figure accordingly in the revised manuscript (now Figure 7).

The methods are not clear on the "Protein trans-splicing assay" e.g. describe the buffer more directly.

Done, thank you.

"Subsequently, Alexa Fluor 488 and Alexa Fluor 647 maleimide (Jena Bioscience), respectively, were added to each of the proteins in three steps"

Usually these dyes are sold by Thermo Fisher and Jena have different dyes?

The mentioned fluorophores were indeed purchased from Jena Bioscience GmbH with the product numbers APC-006-01 and APC-009-1. The products are apparently no longer sold directly by Jena Bioscience but via other distributors. The product name is "AF488 maleimide", for example, but Alexa Fluor® 488 is specified as alternative name. We have changed the specifications to AF488 and AF647 in the revised version of the manuscript.

Stokes, not Stoke, radius

Corrected, thank you.

Fig. S1b: curly arrow at stage 3 not well positioned

Corrected, thank you.

Fig. S8a: include chirality

Included, thank you.

Fig. S9c/d/e: typos in axis labels

Corrected, thank you.

angel instead of angle at various places

Corrected, thank you.

We would like thank all reviewers for their time, efforts and valuable input to help improve this manuscript!